# Benchmarking passive microwave satellite derived freeze/thaw datasets

Annett Bartsch[1], Xaver Muri[1], Markus Hetzenecker[2], Kimmo Rautiainen[3], Helena Bergstedt[1], Jan Wuite[2], Thomas Nagler[2], and Dmitry Nicolsky[4]

[1]b.geos, Industriestrasse 1, 2100 Korneuburg, Austria
[2]ENVEO, Innsbruck, Austria
[3]FMI, Helsinki, Finland
[4]University of Alaska Fairbanks, Fairbanks, 99775, USA

**Correspondence:** Annett Bartsch (annett.bartsch@bgeos.com)

**Abstract.** Satellite derived soil surface state has been identified to be of added value for a wide range of applications. Frozen versus unfrozen conditions are operationally mostly derived using passive microwave (PMW) measurements from various sensors and different frequencies. Products differ thematically as well as in spatial and temporal characteristics. All of them offer only comparably coarse spatial resolution in the order of several km to tens of km which limits their applicability. Quality assessment is usually limited to comparisons with in-situ point records, but a regional benchmarking dataset is thus far missing. Synthetic Aperture Radar (SAR) offers high spatial detail and thus is potentially suitable for assessment of the operational products. Specifically, dual polarized C-band data acquired by Sentinel-1, operating in Interferometric Wide (IW) swath mode with a ground resolution of 5 x 20 m in range and azimuth, provide dense time series in some regions and are therefore suitable as basis for benchmarking. We developed a robust freeze/thaw (FT) detection approach, applying a constant threshold on the combined C-band VV and VH polarization ratios, that is suitable for tundra regions. The achieved performance (91.8%) is similar to previous methods which apply an empirical local threshold on single polarized VV backscatter data.

All global products, tested with the resulting benchmarking dataset, are of value for freeze/thaw retrieval, although differences were found depending on season, in particular during spring and autumn transition. Fusion can improve the representation of thaw and freeze-up, but a multi-purpose applicability cannot be obtained since the transition periods are not fully captured by any of the operational coarse resolution products.

## 1 Introduction

Soil surface state (frozen or unfrozen) can be obtained from space using active and passive microwave (MW) data. Using spaceborne passive microwave data 42% of the northern hemisphere land surface has been identified to be affected by seasonal freezing and thawing (Kim et al., 2012). This includes permafrost regions, where it is a characteristic of the so-called active layer, the upper soil on top of permafrost. Satellite derived freeze/thaw (FT) information is hence suitable as a proxy for permafrost characteristics (extent Park et al. (2016b), ground temperature Kroisleitner et al. (2018)). The performance of passive microwave retrieval algorithms depends on wavelength, acquisition timing, and the algorithm used to derive the surface state

(Johnston et al., 2020). Landcover at sub-grid scale needs to be considered in addition (Bergstedt and Bartsch, 2017; Bergstedt et al., 2020b).

A traditional application area of FT products is the masking of satellite derived soil moisture targeting a separation of periods with and without the presence of liquid water in the upper soil. The requirements are in such cases usually aligned with the soil moisture product (spatial and temporal resolution). However, some of the available products lack consistent FT flagging (Trofaier et al., 2017). Further potential of soil FT products includes for example assessment of the vegetation growing season (Kim et al., 2014, 2020; Park et al., 2016c; Böttcher et al., 2018), evapotranspiration (Zhang et al., 2021; Kim et al., 2018) and fluxes of greenhouse gases (Tenkanen et al., 2021; Erkkilä et al., 2023). The latter includes confining the activity period of soil microbes (Bartsch et al., 2007; Kim et al., 2014) and the combination with greenhouse gas concentrations derived from satellites and permafrost (Park et al., 2016a; Kroisleitner et al., 2018).

Different applications have different needs regarding FT detection. E.g for the climate modelling community a frozen fraction approach is required rather than a frozen/unfrozen classification (Bergstedt et al., 2020b). Subgrid information of snowmelt patterns are also of interest in the context of long-term ecosystem monitoring in the Arctic (Rixen et al., 2022). Surface state products derived by satellites represent either the surface condition of snow and ice or soils. In case of snow the terms 'wet/dry' or 'frozen/melting' are used, while for soil the terms 'frozen/unfrozen' are used to describe the states. Dedicated FT products mostly address soils and thus acquisitions under snow-free conditions, but wet snow information is of added value for such observations due to associated changes in soil temperature beneath the snow pack (e.g. Bartsch et al., 2007; Kroisleitner et al., 2018). Climate change impact can be documented through the length of the unfrozen/snow-free season, but spring snow thaw timing and length of snow melt period needs to be monitored in addition (e.g. Kouki et al., 2019).

45

The production of a FT Climate Data Record (CDR) may require the combination of products from different satellites. Differences between records using different wavelengths have been identified (e.g. Johnston et al., 2020). A range of strategies for fusion of different records have been proposed, specifically L-band passive microwave and C-band scatterometer (Chen et al., 2019; Zhong et al., 2022). Inconsistencies and systematic differences due to acquisition timing and differences regarding wavelength and system (active – passive; Trofaier et al., 2017, static in the beginning, varying towards the end) need further investigation. The utility of the products varies not only based on different user needs but also on the retrieval approach and calibration data. Reanalyses data are commonly used to define a threshold for surface state classification. 0°C is applied as threshold for passive records representing the brightness temperature and reanalysis values relationship through a cosine function (Kim et al., 2014). Naeimi et al. (2012) fit a logistic function to describe a temperature - C-VV backscatter relationship. The turning point of the function is considered as threshold. Kroisleitner et al. (2018) found that the PMW derived frozen period length differs considerably from Naeimi et al. (2012). Possible reasons are the acquisition time, wavelength and calibration approach. These approaches differ from snowmelt products where for example in case of sea ice a temperature above -1°C for 3 consecutive days is considered (Smith et al., 2022). Over land, diurnal thaw and refreeze as condition for

start of spring melt has been suggested alternatively, but this requires higher repeat measurement intervals than usually available (Bartsch et al., 2007). Most products target 80% detection accuracy. A commonly agreed benchmark for this assessment is lacking so far relating partially to lack of representation of spatial variations by the available in-situ data within the footprints.

Consolidated requirements for a FT product aiming at permafrost research have so far only been stated in Bartsch et al. (2022). A 100m spatial resolution product in 10 day intervals has been requested. Kroisleitner et al. (2018, with focus on Metop ASCAT and SSMI) exemplify that daily resolution is required for ground temperature estimation. General requirements for a ground temperature product have been collected as part of the Permafrost_cci baseline project (Bartsch et al., 2023e) which need to be considered in this context. As a threshold requirement for temporal and spatial coverage, at least the last decade needs to be addressed as well as the whole Arctic. The target should be a global product including not only the Northern Hemisphere but also mountain ranges and polar ice free regions on the southern hemisphere. 10 km (threshold) and 1 km (target) resolution respectively are eventually required to serve the needs of permafrost modelling at global scale (Bartsch et al., 2023e).

A major challenge of coarse resolution satellite products from PMWdata across the Arctic is the quality assessment with in-situ data due to the scarce availability of ground stations and loss of representativeness because of high landscape heterogeneity. Available FT products have a nominal spatial resolution in the order of several tens of kilometres. Bergstedt et al. (2020b) demonstrated that considerable variations can occur within a footprint due to topography and/or variability in land surface types. Another option for evaluating the quality of coarse resolution products is by using data based on C-Band which have a higher spatial resolution. So far such investigations have been limited to C-band scatterometer, specifically Metop Advanced Scatterometer (ASCAT).

A strategy for a SAR-based benchmark dataset creation which relies on calibration with reanalyses data is described in Bergstedt et al. (2020b). This calibration approach follows the methodology proposed for scatterometer by Naeimi et al. (2012). It needs to be carried out for each footprint/pixel individually since various factors, such as surface roughness and volume scattering in the surface layer, may impact the absolute backscatter intensity values. Naeimi et al. (2012) compared gridded temperature data (ERA-Interim reanalyses data) and Metop ASCAT C-Band VV backscatter data (grid spacing appr. 12.5km) and fitted a logistic curve. The lower flat part of the function represents frozen conditions, while the upper flat part relates to the unfrozen conditions. The inflection point provides the location specific threshold. This method requires suitable temperature data for each grid point. As this is not available at high spatial resolution as provided by Sentinel-1 (5 x 20 m; 10m nominal resolution) a method independent from such parameterization needs to be developed for regional applications. An algorithm which for example makes use of a universal threshold is required. The study by Bergstedt et al. (2020b) was in addition limited to VV polarization matching ASCAT properties. Backscatter intensity expressed as $\sigma_0$ was used and incidence angle normalization to 40 degree was applied using the approach by Widhalm et al. (2018) in order to match ASCAT specifications. An overall accuracy of 94% was obtained in comparison to soil temperature data from a tundra area. Missions such as Copernicus Sentinel-1 operate in dual polarization mode, providing co- and cross-polarization over most land areas (commonly VV and

VH). The use of both for FT has been investigated only in very few cases so far. Cohen et al. (2021) calculate the backscatter difference (of each to be classified date) with respect to a frozen and unfrozen reference (pre-selected representative scenes). The study was eventually also limited to VV as the focus was on forested environments where VH is expected to be influenced by the canopy. $\sigma_0$ was used and ground and canopy contribution separated (including consideration of incidence angle). A static, but regional specific threshold applied to the difference between the deviation from the frozen and thawed reference respectively is suggested. Similarities with air temperature was between 64% and 99% depending on region. A further approach which has been regionally tested for Sentinel-1 is the use of a convolutional neural network (CNN) algorithm trained also using backscatter from a frozen and unfrozen reference period, using both VV and VH, plus incidence angle and landcover information (Chen et al., 2024). A classification accuracy of 88% was obtained in comparison to soil temperature data across an area north of the treeline where the CNN was trained.

Algorithms exist which use backscatter ratios for wet/dry snow detection based on Sentinel-1. This allows the application of a global threshold value to distinguish between states without a location specific parameterization also avoiding the need to directly consider the incidence angle (Nagler and Rott, 2000). The scheme has so far not been tested for soil FT. Previous calibration and validation activities have found that a global threshold of -2 dB (ratio between acquisition and reference image) is suitable to discriminate wet and dry snow. It is hypothesized that a similar strategy is suitable for frozen versus unfrozen soil classification. This requires sites with good in-situ data availability which are scarce in permafrost regions. A comparably high number of borehole records is, however, available over the Alaskan North Slope (Biskaborn et al., 2019). Also, measurements from a FT dedicated sensor network are available from Northern Finland (Bergstedt et al., 2020b).

The purpose of this study is to evaluate the utility of C-band SAR data available at VV and VH polarizations for benchmarking of passive microwave satellite derived freeze/thaw products which have coarse resolution but global coverage. In-situ air as well as soil temperature from different locations across the Alaskan North Slope are used to identify a suitable ratio threshold for the benchmark dataset creation. Validation includes near surface soil temperature records from Northern Finland. The focus is on tundra areas and the evolution of thaw and freeze-up within the footprint of the coarse resolution products and the discussion of implications of differences between the datasets for a range of potential applications.

## 2   Data and analyses regions

### 2.1   Sentinel-1

The Copernicus Sentinel-1 C-band SAR mission consisted of two satellites for the analysis period 2014 to 2020. Sentinel-1A was launched in April 2014 and Sentinel-1B followed in April 2016. In order to provide global coverage, the Sentinel-1 mission is operating in a 12 days repeat cycle over most land surfaces.

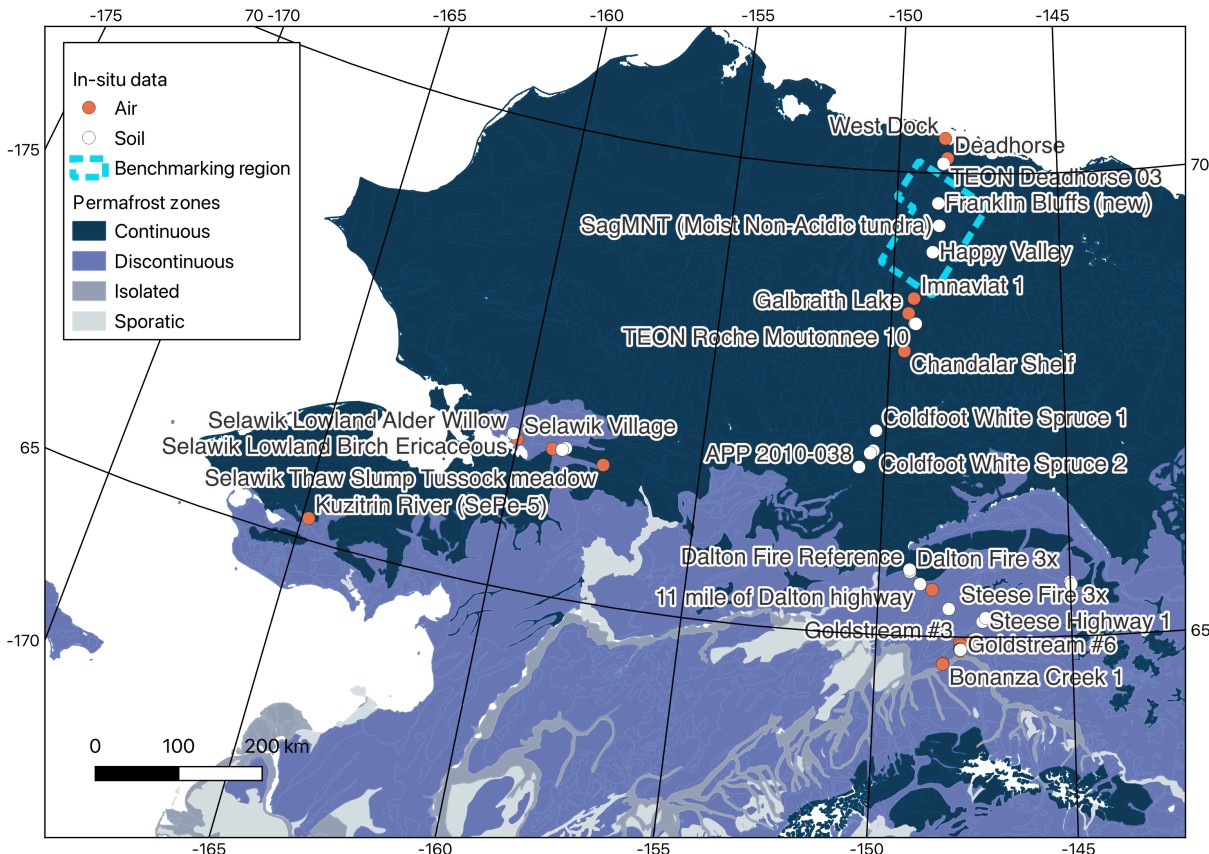

**Figure 1.** Location of in-situ data sites with air (red) and soil (white) temperature for 2018 to 2020 (sources: Romanovsky et al. (2020, 2021, 2022)). Background: permafrost zones from Jorgenson et al. (2008), from dark to light blue: continuous, discontinuous, sporadic, isolated. Dashed outline indicates region for benchmarking and fusion.

The capability of Copernicus Sentinel-1 for FT products has been demonstrated e.g. by Bergstedt et al. (2020b) and Cohen et al. (2021). The primary data source for the benchmarking is therefore Sentinel-1 C-band SAR data acquired in IW (Interferometric Wide swath) Terrain Observation with Progressive Scans (TOPS) mode at VV and VH polarization. Acquisition types and coverage vary considerably across the Arctic (Bartsch et al., 2021a). The required data are unavailable for Greenland and the Canadian High Arctic (Bartsch et al., 2021b). Comparably good coverage is available for Alaska as well as Scandinavia.

These areas overlap with in-situ measurement sites relevant for this study. All scenes starting 30th of July 2017 to 31st of December 2020 have been processed for northern Alaska (covering all automatic weather stations and borehole locations, see Figure 1) and all scenes for August 2016 to August 2018 covering the Kaldoaivi area in Northern Finland (Figure 2).

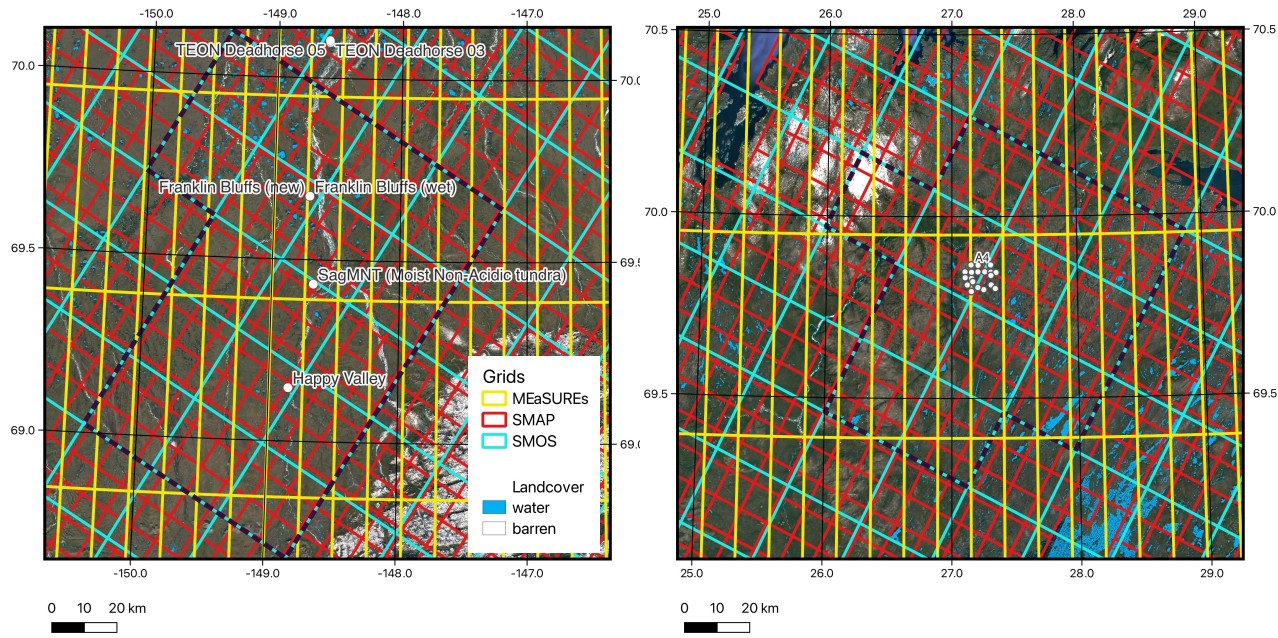

**Figure 2.** Examples of satellite product grids (see Table 1) for sites with in-situ borehole data on the Alaskan North Slope (left) and iButton data at Kaldoaivi, Northern Finland (right). Background: selected landcover classes relevant for masking (source: Bartsch et al. (2023c)) and Google hybrid. in-situ sites shown as white filled circles. Black dashed lines indicate the boundary of SMOS cell extent used for assessment and fusion. Grids of the passive microwave products (MEaSUREs, SMAP and SMOS) differ in projection and extent. For details see Table 1.

## 2.2 Global/northern hemisphere products

Three FT datasets based on relatively coarse spatial resolution PMW information are considered for benchmarking (Table
1). The currently listed FT Fundamental Climate Data Record (FCDR) in the catalog of CEOS (Committee on Earth Observation Satellites) Working Group on Climate is the MEaSUREs (Making Earth System Data Records for Use in Research Environments) Global Record of Daily Landscape Freeze/Thaw Status, Version 3 (Kim et al., 2014) which is associated to the ECV (Essential Climate Variable) soil moisture. It is based on various passive microwave missions (SMMR, SSM/I, and AMSR-E/AMSR2 (referred to as AMSR in the following)) and goes back to 1979. It is provided with 25km
gridding, separation of AM and PM, and >80% mean annual spatial classification accuracy. FT information is supplied binary (either frozen or unfrozen). Vegetation related use of the product is suggested apart from soil moisture masking (https://climatemonitoring.info/ecvinventory/). An updated version (version 5) which is available through the National Snow and Ice Data Center (NSIDC; Kim et al. (2017, 2021)) was used for this study. Version 5 is also considered for directly deriving landsurface temperature for the production of climate records within the European Space Agency CCI LST project (Climate
Change Initiative Land Surface Temperature, Dodd et al. (2021)). The AMSR time series of the MEaSUREs dataset is available

**Table 1.** Freeze/thaw datasets with northern hemisphere coverage considered for benchmarking with Sentinel-1 retrievals. V - vertical, H - horizontal, NPR - Normalized Polarization Ratio, SCV - single channel vertical polarization.

| Dataset, Mission(s) | Scheme | | Start year | Used gridding in km | Used Equal-Area Scalable Earth (EASE)-Grid 2.0 version | Used flags |
|---|---|---|---|---|---|---|
| | Frequency and polarization | Method | | | | |
| MEaSUREs v5; SMMR, SSM/I, and AMSR | 36 or 37 GHz, V polarization | Seasonal threshold, single band; Separate products for SMMR, SSM/I, and AMSR | 1979 | 25 | Global (cylindrical, equal-area) | frozen, unfrozen (AM) |
| SMAP | 1.41 GHz, H and V polarization | Seasonal threshold, NPR or SCV and AMSR-E mitigation | 2015 | 9 | Northern Hemisphere (Lambert azimuthal equal-area) | frozen, unfrozen (AM) |
| SMOS | 1.41 GHz, H and V polarization | Two thresholds determined from extreme values, scaled NPR | 2010 | 25 | | frozen, unfrozen, thawing |

as 6 km resolution record in addition to the 25 km record which includes time series from all sensors. The 6km dataset covers the northern and southern hemisphere currently spanning the period 2002-2021 (Kim et al., 2021). However, only the 25 km product (consistent with SSM/I) was used for this study. AM data was chosen where available since most available SAR data for the Alaskan benchmarking site (97%) were acquired at AM times.


Further products exist from SMAP (2015-) and SMOS (2010-) and are available on an operational basis (Kraatz et al., 2018; Rautiainen et al., 2016). Both are L-band PMW missions and provide therefore information on the upper centimeters of the soil. The SMAP mission specifically targets freeze/thaw, complementing soil moisture, and requirements have been defined for the mission (Dunbar, 2018; Entekhabi et al., 2014). They have been phrased for the radar instrument (which is currently not

in operation). Surface binary freeze/thaw state in the region north of 45ºN latitude, which includes the boreal forest zone, was targeted with a classification accuracy of 80% at 3 km spatial resolution and 2-day average intervals. A two-day precision at the spatial scale of landscape variability ($\approx$ 3 km) was originally targeted. The available SMAP PMW product considers four stages: frozen, thawed, transitional, inverse transitional (Derksen et al., 2017). The two transitional categories refer to days with differences between AM and PM acquisitions. The SMAP product shows >70% accuracy for the 36 km grid version according

to Kraatz et al. (2018). 78 and 90 % accuracy were determined for descending (AM) and ascending (PM) orbit observations respectively in relation to independent surface air temperature-based FT estimates from $\sim$ 5000 global weather stations (Kim et al., 2019). The SMOS mission FT product addresses three stages: thawed, partially frozen and frozen covering the northern

hemisphere (daily, 25 km; Rautiainen et al. (2016)).

The different sensors represent different frequencies (see Table 1) and different strategies were followed for the FT retrieval. The MEaSUREs algorithm is based on a seasonal threshold approach. The threshold was derived annually on a grid-cell-wise basis (Kim et al., 2017) using an empirical relationship between brightness temperature and daily surface air temperature records from global model reanalysis. The SMMR-SSM/I-SSMIS record was developed by merging the Scanning Multichannel Microwave Radiometer (SMMR), Special Sensor Microwave Imager (SSM/I), and Special Sensor Microwave Imager/Sounder

(SSMIS) 37 GHz frequency (vertical polarization) brightness temperature records (Kim et al., 2014). The AMSR records represent 36 GHz. The global 25 km datasets are provided as a cylindrical, equal-area projection, what translates to 10 km x 62.5 km at the latitude of the benchmarking sites (Figure 2).

    SMAP FT state was determined using a seasonal threshold approach (Derksen et al., 2017). A normalized polarization ratio

(using V and H brightness temperature; 1.41 GHz) is assessed. A winter and summer reference is derived from frozen and thawed soil conditions. It is calculated for each year and averaged over the entire SMAP period (according to the documentation of Xu et al. (2023)). These values are used for derivation of a threshold and a seasonal scale factor. In cases of low SCV (single channel vertical polarization) correlation with physical surface temperature, SCV retrieval is used and in some cases an AMSR-E mitigation scheme applied (Xu et al., 2023). The dataset is provided in Lambert azimuthal equal-area projection and

available at 36 km and 9 km (enhanced level 3) gridding. Only the latter was used.

    Thaw and freeze references are also used in case of SMOS FT retrieval (Rautiainen et al., 2016; Rautiainen and Holmberg, 2023). For FT state estimation, the normalized polarization ratio (NPR, 1.400 GHz–1.427 GHz) values are scaled using the reference NPR values from frozen and thawed soil conditions. All potential observations from the frozen soil and thawed soil

conditions are collected from the winter and summer periods when reanalysis air temperature data has indicated <-3C and >+3C, respectively. From these collected potential values, the 50 most extreme ones are stored and their median used as a reference value. These are pixel-wise values. Then two thresholds are used to differentiate between three FT states from the scaled NPR. The dataset is provided in Lambert azimuthal equal-area projection and 25 km gridding.

Several active microwave products have been published previously but are limited temporally and spatially and are thus not considered for benchmarking. These include an experimental product from Metop ASCAT (C-band scatterometer, 12.5 km gridding, Paulik et al. (2012)), starting in 2007, aimed at permafrost and flux applications (Naeimi et al., 2012). It is available for 12.5 km nominal resolution and considers three states: frozen, unfrozen, melting. In addition, a C-band SAR product at 1 km was developed based on ENVISAT ASAR GM (Park et al., 2011) and used to assess the ASCAT product (Bergstedt and

Bartsch, 2017).

**Table 2.** In-situ datasets used for calibration and validation of the benchmark dataset creation, as well as for the fusion assessment. AWS (Automatic Weather Station) and borehole record sources: Romanovsky et al. (2020, 2021, 2022).

| Region | Instrumen-tation type | Data type | data points | time period | Use |
|---|---|---|---|---|---|
| **Alaska, including North Slope** | AWS | Air temperature | 22 | June 2018 - December 2020 | calibration |
| **Alaska, including North Slope** | boreholes | soil temperature (closest to surface: 1-75cm, median 22 cm) | 26 | June 2018 - December 2020 | calibration (all), fusion assessment with subset (three sites with sensors at 5-7 cm |
| **Northern Finland, including Kaldoaivi** | iButtons | near surface soil temperature (2-3cm) | 24 | August 2016 - August 2018 | validation |

Benchmarking of the PMW datasets has been carried out over 17 SMOS grid cells (and overlapping SMAP and MEaSUREs cells) for Alaska and 12 cells for Northern Finland (Figure 2).

## 2.3 Temperature records

Temperature records from in-situ observations include long-term measurement sites such as boreholes and automatic weather stations (AWS) as well as short-term and spatially distributed observations using iButton data loggers. The latter network was established for validating surface conditions derived from C-band satellite data (for details see Bergstedt et al. (2020b)). They have been used for inter-comparing Sentinel-1 with a coarser regional dataset based on Metop ASCAT covering the region Kaldoaivi in Northern Finland. This site is located within a palsa mire and represents a zone of sporadic permafrost. The temperature sensors have been distributed in the proximity of a permafrost borehole site. The majority of the iButtons are located outside of permafrost affected landscape elements. The snow water equivalent (SWE) varies across the sites in winter, but at most sites snow depth is sufficient for insulation and temperatures remain close to 0°C below the snowpack. The available record includes 24 sites spanning two years (August 2016 to August 2018). The iButton loggers were placed approximately 2–3 cm below the surface to avoid direct warming influence by the Sun.

Records from boreholes and automatic weather stations (AWS) across Alaska (Romanovsky et al., 2020, 2021, 2022) including the North Slope, representing continuous to discontinuous permafrost, have been used for calibration (Figure 1). Data is available from 26 boreholes with depths varying between 1-75 cm and from 22 AWS providing air temperature. Three years of data (2018 to 2020) have been used. ERA5 reanalyses air temperature data for the same time period were used in addition for the evaluation of the fused dataset to allow comparability with previous studies.

## 3    Methods

### 3.1    General workflow

First, the benchmarking dataset was created using Sentinel-1. The implementation of surface state classification building on intermediate products of the wet/snow retrieval as proposed in Nagler and Rott (2000) requires a threshold determination in a first step. For calibration in-situ data from Alaska limited to the upper few centimeters of the soil have been used. The target landcover type (as determined by the potential applications of a FT product) are soils. Water surfaces, bare areas and glaciers therefore need to be masked with a dataset of similar spatial resolution (Bergstedt et al., 2020b). The circumpolar landcover units by Bartsch et al. (2023d) fulfill this requirement and have therefore been used for masking.

Eventually, the frozen fraction is derived individually (spatial subsetting) from the masked data for each cell of the PMW FT products listed in Table 1. The resulting time series are used for the evaluation and the development of the fusion scheme. Fused records are assessed with the benchmark dataset similarly as the input records as well as with in-situ records. The fused record examples are combined with a dataset providing mid-winter thaw and refreeze of snow (Bartsch et al., 2023a), where such events occur, to facilitate the discussion.

### 3.2    Pre-processing of SAR data

The original wet/dry snow detection utilizes repeat pass dual-pol Copernicus Sentinel-1 SAR data (acquired in IW swath mode) from different tracks. The method includes the generation of reference images, applying a statistical pixel-wise analysis of tens of co-registered repeat pass Sentinel-1 images (Nagler et al., 2016). To exploit the dual-polarisation capabilities of Sentinel-1 the backscatter ratio of the snow covered period and reference data are calculated separately for co- and cross-polarisation and then combined. The combination accounts for differences in angular backscatter behaviour of co- and cross-polarized backscatter data by applying the local incidence angle as weighting factor. The result is a weighted ratio combination $R_C$ (Nagler et al., 2016) based on the backscatter ratios of VV ($R_{VV}$) and VH ($R_{VH}$).

$$R_C = W R_{VH} + (1 - W) R_{VV} \tag{1}$$

The weight ($W$) varies with the local incidence angle ($\theta$) applying the following rule:

$$
\begin{aligned}
IF(\theta < \theta_1) &\rightarrow \{W = 1.0\}, \\
IF(\theta_1 \leq \theta \leq \theta_2) \rightarrow \left\{W = k \left[1 + \frac{(\theta_2 - \theta)}{(\theta_2 - \theta_1)}\right] \right\}&, \\
IF(\theta > \theta_2) &\rightarrow \{W = k\}.
\end{aligned}
\tag{2}
$$

For the data sets in this study, we use $k = 0.5$, $\theta_1 = 20°, \theta_2 = 45°$ following Nagler et al. (2016). The SAR data have been sampled to 100mx100m.

More than 1900 scenes have been pre-processed in total over both regions. In-situ locations in Alaska were covered by 52 to 179 scenes each. 115 scenes have been used over Finland for comparison with in-situ records. The time period covered for Finland corresponds with the in-situ data availability (August 2016 to August 2018, acquisitions every 12 days). Data processed for Alaska span three years from 2018 to 2020 (on average an acquisition every 9 days using overlapping orbits). This results in differing sampling intervals between the regions and lower sampling compared to the global FT records.

### 3.3 Freeze/thaw retrieval from SAR data and evaluation

The last step of the wet snow retrieval scheme of Nagler et al. (2016) is the segmentation of wet snow versus snow-free and dry snow areas based on the $R_C$ images (Nagler et al., 2016). This step was adjusted for the FT retrieval. Specifically, a suitable threshold $Thr$ needed to be defined for the segmentation of frozen versus unfrozen soil.

The threshold determination followed the approach originally suggested for VV backscatter time series by Naeimi et al. (2012) (Scatterometer) and adapted by Bergstedt et al. (2020b) (SAR). The threshold was defined based on a logistic function fit between backscatter and temperature data and the threshold $Thr$ determined from the inflection point. This was based on reanalyses data in Naeimi et al. (2012). Bergstedt et al. (2020b) tested calibration based on in-situ air temperature over a limited number of sites and validated the results with in-situ near surface soil temperature data. Here we also used in-situ data, air and soil, but from a larger dataset and also for calibration. Air as well as soil temperature time series were compared to the $R_C$ value of each corresponding 100mx100m SAR pixel. The inflection point of the fitted function was derived for both types and the corresponding backscatter value used as threshold to distinguish between frozen ($\leq Thr$) and unfrozen ($> Thr$) conditions. Here, we applied it to the combined VV and VH ratio values ($R_C$) as defined in Nagler et al. (2016). The logistic function was defined based on the processed $R_C$ data on the Alaskan North Slope. The fitting procedure included the consideration of the maximum value of $R_C$ and the average for all values acquired under conditions below -5°C. For the borehole records the upper most sensor below the ground was selected. Validation of the threshold approach was carried out based on the near surface temperature observations available for Northern Finland (Kaldoaivi). For this purpose, daily averages of 0°C and lower were defined as frozen and all values above as unfrozen.

A disadvantage of using backscatter intensity at C-VV is the temperature dependence at cold conditions (Naeimi et al., 2012; Bergstedt et al., 2020b; Bartsch et al., 2023b). The potential influence was therefore also investigated using the available in-situ records. In addition to $R_C$, C-VV and C-VH backscatter intensity input values were compared with the temperature measurements for this purpose.

### 3.4 Sub-setting global and fused dataset records for evaluation

The available global products represent varying grids. Therefore, the Sentinel-1 backscatter ratio analyses needed to be carried out over sufficiently large areas in the proximity of the in-situ locations (overlap of grids, Figure 2). The benchmarking extent was chosen considering in-situ data availability as well as distance from coast in order to limit potential quality issues in the PMW datasets. Both sites are labeled as high quality areas in the MEaSUREs dataset. SMOS quality flags indicate temporarily up to 5 (out of the 17) cells as potentially affected by radio frequency interference (RFI) over northern Finland. No RFI issues are listed for the Alaskan sites. SMAP flags indicate retrieval with SCV (single channel vertical polarization) due to correlation <5% with temperature data and in some cases the use of AMSR or TB (brightness temperature) mitigation. The latter occurred only outside of the transition periods, in mid-summer and mid-winter (Figure A1). For Alaska this includes mostly end of December to end of March, and in some cases also July and August. For Finland flagging occurred from July to September (where freeze-up starts in October).

Permanent snow and ice covered areas, water and barren (bedrock) have been extracted from landcover for the masking of 'none-soil' in the SAR images. The 'none-soil' fraction can be very large in permafrost environments. Figure 2 shows lakes and bare area as derived from a landcover map (Bartsch et al., 2023c). Both landcover types were excluded from the frozen fraction calculations.

The frozen/unfrozen fraction was derived for each of the overlapping footprints of the PMW products (grid cells as shown in Figure 2). The proportion of unmasked Sentinel-1 pixels classified as frozen has been determined for each acquisition. Basic statistics were derived for each global product separately (including for the presentation of results as violin plots).

### 3.5 Fusion: scheme and evaluation

Criteria for preference of a certain dataset were eventually defined for a fusion scheme based on the benchmarking. The year is split into three phases for this purpose: DOY (day of year) 1 - 209, 210-300, and >300. In a first step, dataset(s) to be considered for a certain period were selected. In a second step the conditions were defined (considering agreement between datasets/ results of benchmarking). The combined dataset (referred to as Fusion) was eventually also compared to the benchmark dataset derived from the Sentinel-1 frozen fraction. The fusion of the daily records was made on the basis of the SMOS grid (25km). All comparisons of the Fusion dataset were also made on the basis of the SMOS grid. The 'partially frozen' flag was separately analyzed from the other flags ('thawed' and 'frozen').

Data from three borehole locations on the Alaskan North Slope were used for the assessment of the unfrozen and frozen flags (temperature sensors at 5-7cm depth) of the fusion product. In addition, ERA5 reanalysis data (air temperature) were investigated at the same sites (for location see Figure 2). The frozen state retrieval from the soil and ERA5 air temperature data was based on the following daily mean temperature thresholds: values <-1°C are considered as frozen and values >1°C as

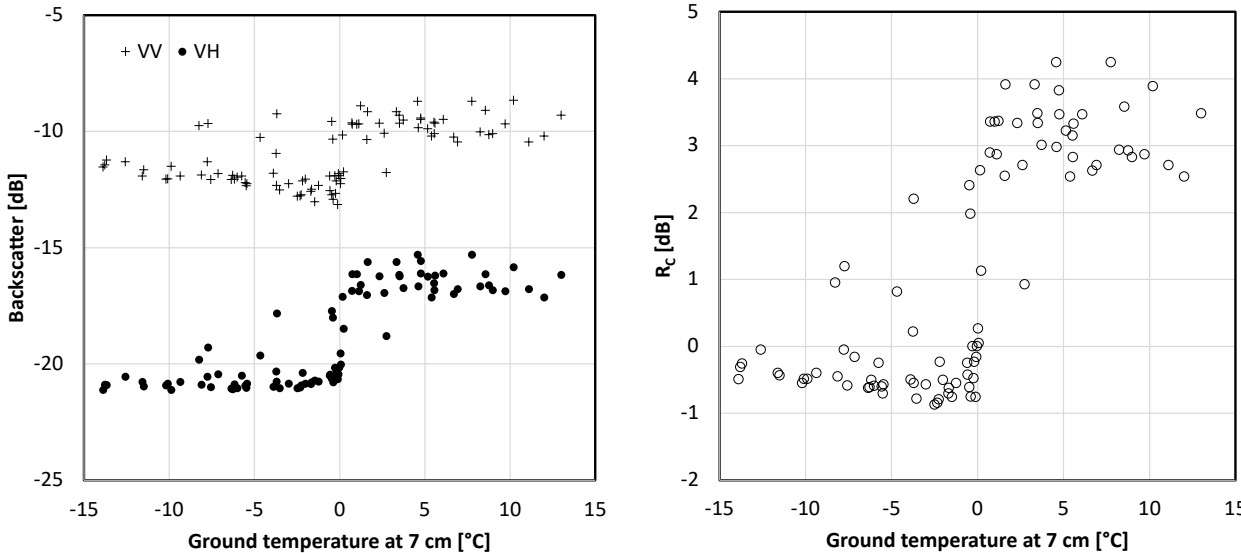

**Figure 3.** Sentinel-1 backscatter intensity at VV and VH (left) and combined ratios $R_C$ (right) (92 acquisitions) versus daily sub-ground temperature (7 cm depth, 5 June 2018 to 27 December 2020) at Sagwon (SagMNT, Alaska). For location see Figure 1).

unfrozen, while values in between are classified as partial thaw.

Days with mid-winter thaw and refreeze of the snowpack or wet snow were not part of the benchmarking but can also be potentially added using existing products. As an example, mid-winter thaw and refreeze as available from combination of Metop ASCAT and SMOS (Bartsch et al., 2023b) has been added and an own flag value assigned. Such data are applicable for November to February (day of year 300 to 60).

## 4 Results

### 4.1 Temperature effect

The temperature influence on C-VV backscatter as described in Bergstedt et al. (2020b) and Bartsch et al. (2023b) does not apply to VH (Figure 3). Backscatter remains at a similar level for all temperatures below 0°C at this polarization combination. Differences between the years due to changes in snow structure, as can especially occur due to rain-on-snow events (Bartsch
et al., 2023b), are also less pronounced at VH. The temperature influence is visible in the combined VV and VH ratio values ($R_C$ increasing with decreasing temperature), inherited from VV, but the magnitude is lower than in the VV input data.

## 4.2 Sentinel-1 threshold determination and validation

The determined value of the inflection point from the logistic function fitting varies across sites (examples for comparison with soil temperature in Figure 4). The spread of the ratio values in winter as well as summer is large, in the order of one to two, but they are largely distinct.

Only minor differences are found between results using soil temperature and air temperature data (Figure 5). The mean values of the inflection point differ by only 0.12 dB. The range is higher for air temperature. All values are higher than the suggested threshold for wet snow (-2 dB, Nagler et al. (2016)). Most inflection point values even exceed it by more than 3 dB. A threshold of 1.34 based on the soil data (median) is suggested for regions with a soil layer that can contain water/ice (basic requirement for FT detection).

The applicability of the threshold could be confirmed using independent records, by use of the in-situ data from Northern Finland (Kaldoaivi region, see Table 2). In total more than 1800 samples of daily average temperature from 24 sites have been available for the Kaldoaivi region. 93.2% of all samples within the two year time period were correctly classified for unfrozen conditions, and 91.2% of the frozen conditions (average 91.8%).

## 4.3 Freeze/Thaw fraction within PMW grids

Only grid cells of the PMW datasets that are completely covered by the Sentinel-1 scenes have been cross-compared (Alaska: Figure 6; Finland: Figure 7). The shown results are limited to two periods: April to June (from completely frozen to completely unfrozen) and August to November (from completely unfrozen to completely frozen) in order to identify potential issues for the different transition types. Violin plots have been chosen to visualize the data distribution (of frozen fraction) for thawed and frozen conditions as defined in the products. For SMOS, the fraction was also analysed for the 'thawing' flag which represents partially unfrozen conditions. Time series examples are provided for two distinct borehole locations, Happy Valley and Sagwon (Figure 8). The frozen flags of each product were translated into values of 10 (unfrozen), 20 (partially frozen) and 30 (frozen) for comparability in these cases. Near surface soil temperature and frozen fraction are shown together with the new flag values. In spring, the frozen fraction as detected by Sentinel-1 drops when the in-situ temperature (1 cm depth) starts to rise above the late winter level at Happy Valley. This drop does not coincide with the beginning of the thaw in the PMW cells. The start of frozen fraction increase does, however, agree with the drop below 0°C in the 2020 example. This is similar for the second site (Sagwon, 8 cm depth, Figure 8). Both borehole records show the so called zero curtain effect at freeze-up. Temperatures can remain at almost 0°C for weeks to months due to latent heat which maintains the temperatures (Outcalt et al., 1990). The frozen fraction is already at 100% weeks before the ground temperature eventually lowers.

The fraction comparison clearly shows a frozen state detection issue for SMAP during the winter on the Alaskan North Slope (Figure 6). A higher number of grid cells is defined as unfrozen than for the other datasets. This effect is more pronounced for April to June than for August to November. This leads to a premature thaw detection in spring. The timing of complete freeze-up is, however, in general captured when the flag switches from unfrozen to frozen as can be demonstrated for the Happy

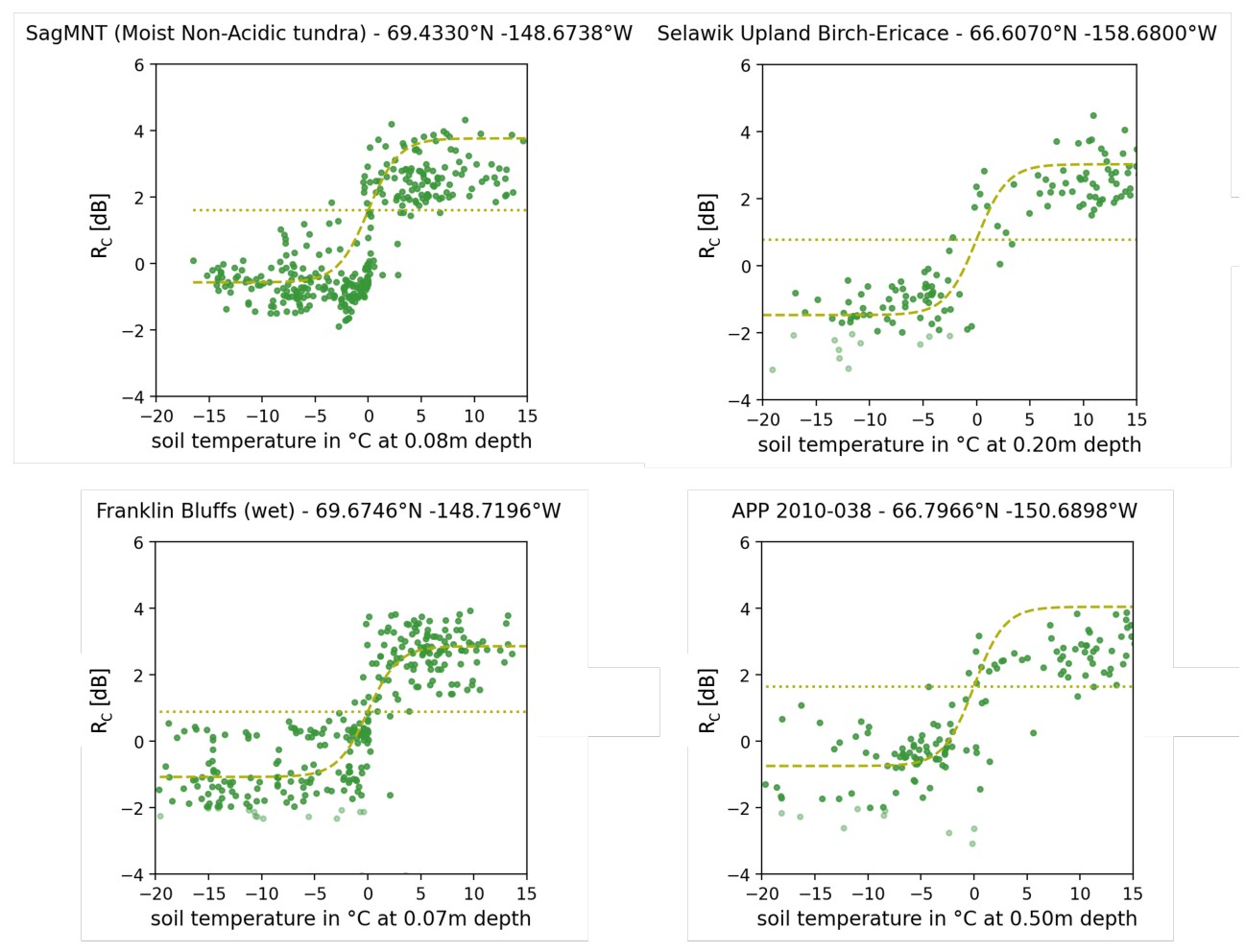

**Figure 4.** Examples for threshold determination at four borehole sites. Sentinel-1 combined polarization ratios ($R_C$) and in-situ soil temperature are compared. Light green indicates $R_C$ values below the wet/dry snow threshold (as defined in Nagler et al. (2016)) Dashed line – fitted function. Dotted line – determined inflection point. For location of sites, see Figure 1).

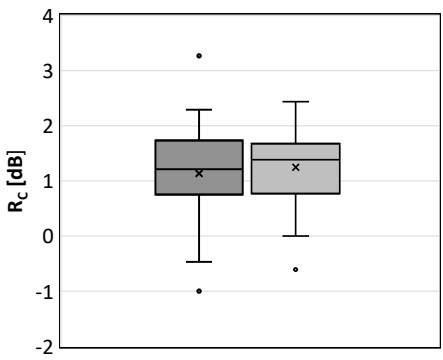

**Figure 5.** Sentinel-1 combined polarization ratios ($R_C$) statistics (boxplots) of inflection points of the logistical function fitted to temperature data following the threshold detection method of Naeimi et al. (2012) (air and soil temperature, 26 and 22 samples respectively; in-situ record overview in Figure 1).

Valley borehole location for 2020 (Figure 8). The unfrozen flag is assigned to the SMAP grid cell from the beginning of the depicted time period, starting 1st of April 2020. The frozen fraction as determined by Sentinel-1 drops below 100% towards the end of April and reaches 0% in June. The 'unfrozen flag' issue at Happy Valley returns mid November. This happens only occasionally at the Sagwon site (Figure 8). AMSR and SSMI flags switch from frozen to unfrozen by end of May when the Sentinel-1 frozen fraction is at almost 0%. The 'partially frozen' flag in SMOS only occurs here in autumn (coinciding with a

frozen fraction of about 80%) what can also be shown for a further borehole location for autumn transition in 2019 (Sagwon, Figure 8) and agrees with the fraction comparison results from the North Slope (Figure 6).

SMOS captures the mid to finalization of thaw on the North Slope well and is closer to the start of thaw for Northern Finland (unfrozen flag includes a higher fraction of frozen conditions, Figure 7). Freeze-up is detected with a delay. The majority of

360 days with partially thawed flag show almost complete freeze-up (spatially) in Sentinel-1. The MEaSUREs AMSR and SSMI products largely agree with each other (time series examples Figure 8). The spring switch occurs at start to mid-thaw at the Happy Valley borehole location. The average frozen fraction with unfrozen flag in spring is lower than for the other products.

The benchmarking results suggest the need for a fusion of all available products (Table 3). This allows a start of the time series in 2015 only (SMAP launch). The start and end of thaw can be represented back to 2010 (SMOS). The MEaSUREs

records back to 1979 allow good detection of the start to mid thaw.

## 4.4 Suggested fusion scheme based on the benchmarking

SMOS and MEaSUREs (SSMI or AMSR) are recommended to be considered in all three time periods (Table 4). The performance of these products was shown to be similar for early winter, therefore either of the products can be used. For late winter

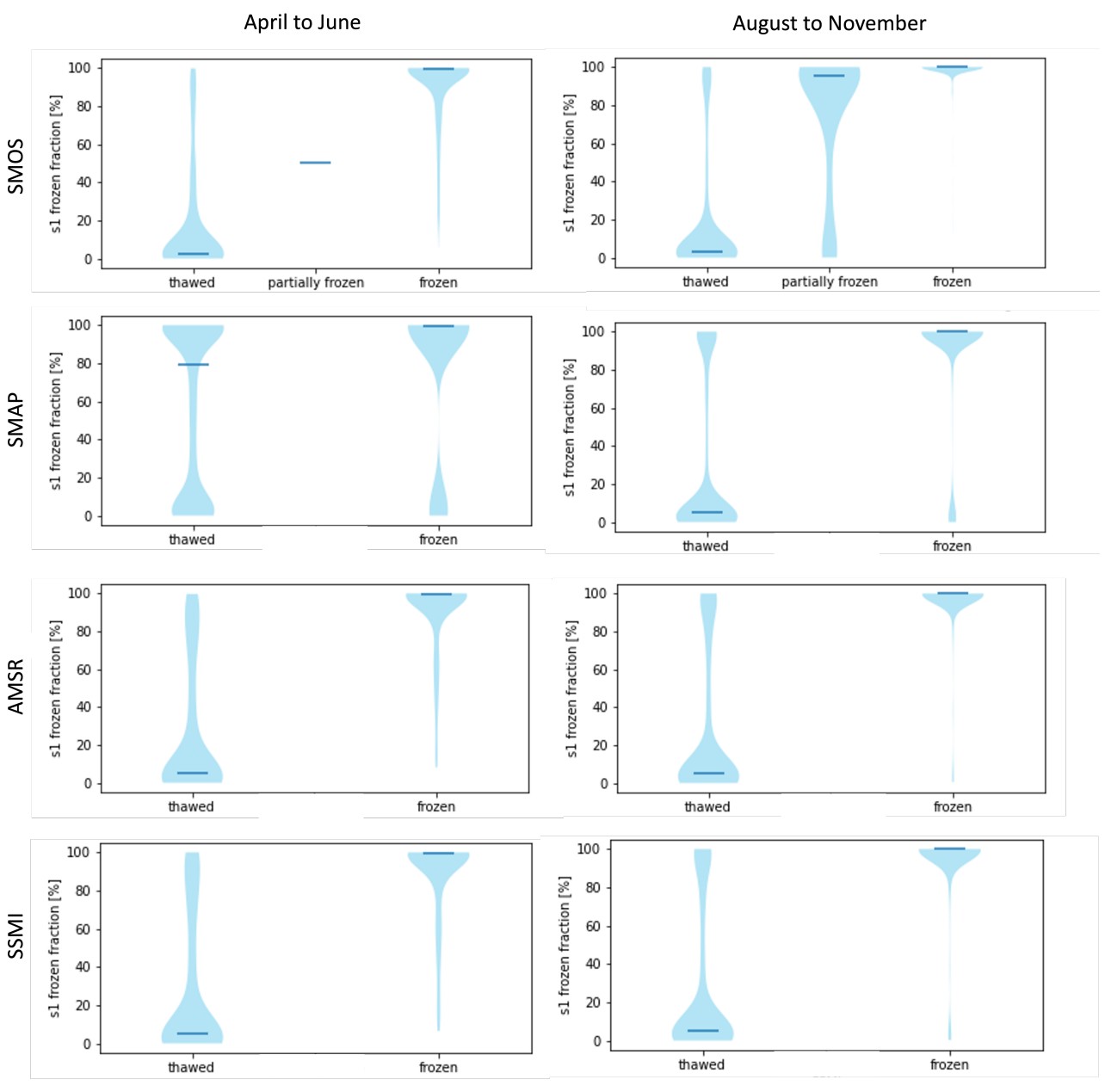

**Figure 6.** Sentinel-1 (s1) derived frozen fraction for footprints of SMOS, SMAP and MEaSUREs (AMSR and SSMI) freeze/thaw products. Sample period 2017 to 2020, Alaskan North Slope. Horizontal lines indicate average values. For analyses extent see Figure 2.

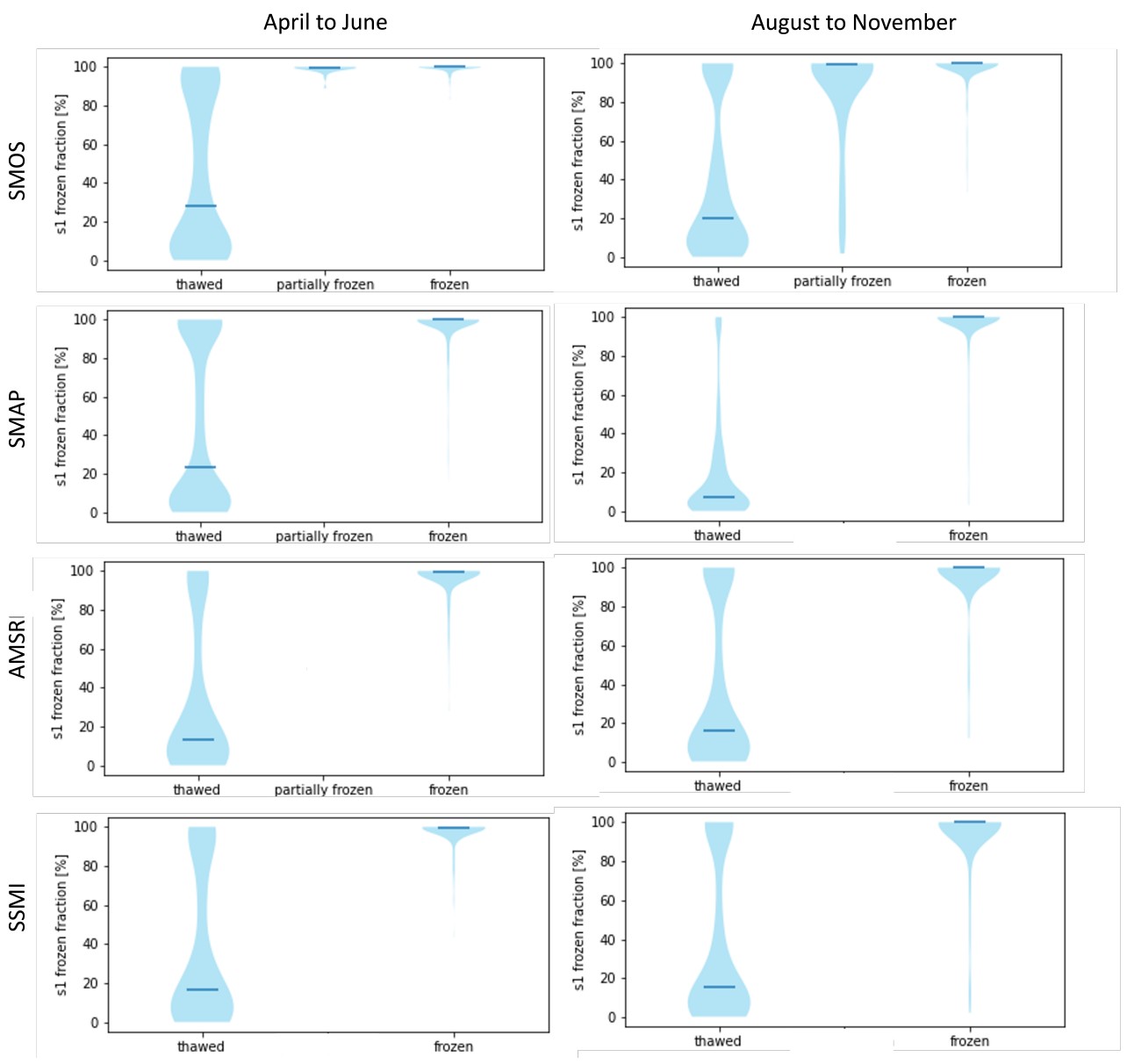

**Figure 7.** Sentinel-1 (s1) derived frozen fraction for footprints of SMOS, SMAP and MEaSUREs (AMSR and SSMI) freeze/thaw products. Sample period 2018 to 2020, Kaldoaivi, Northern Finland. Horizontal lines indicate average values. For analyses extent see Figure 2.

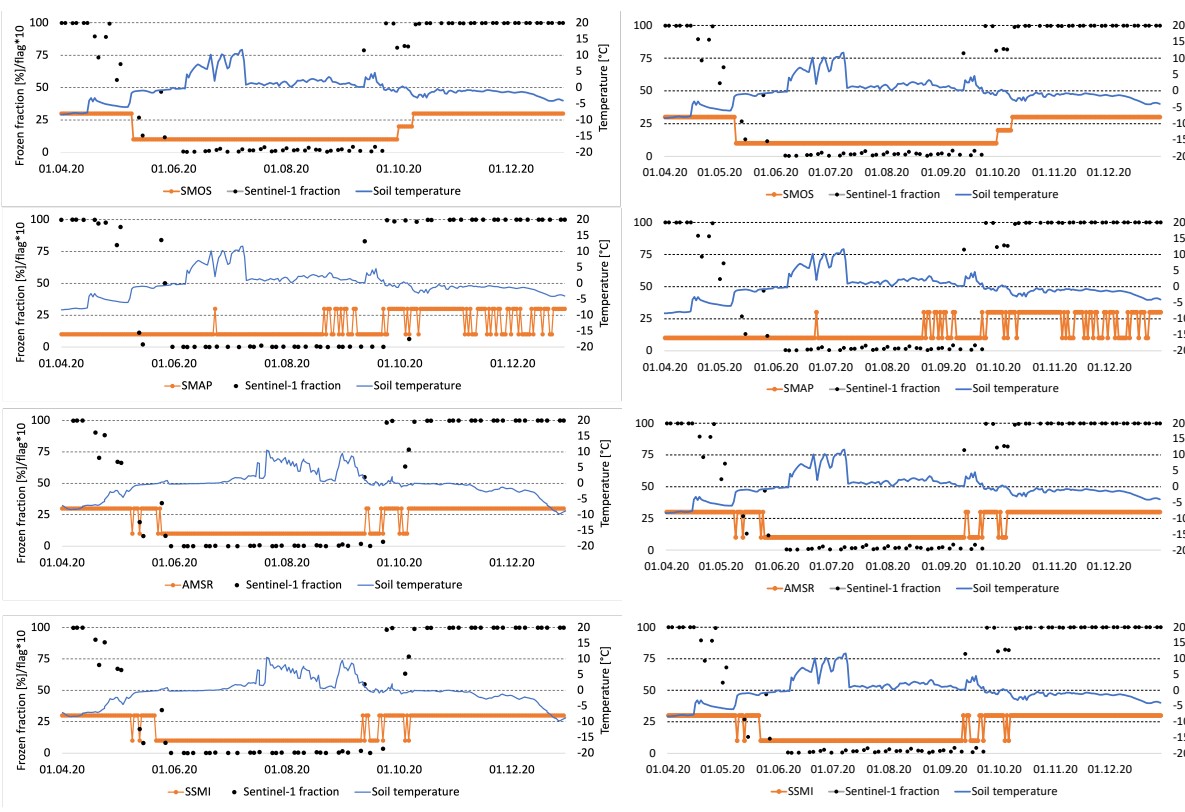

**Figure 8.** Time series for grid cells of passive microwave (PMW) freeze/thaw products (see Table 1) overlapping with the Happy Valley borehole location for 2020 (left) and the Sagwon (SagMNT) borehole location for 2019 (right). Near surface soil temperature (1 cm and 8 cm respectively), Sentinel-1 frozen fraction (differs between PMW datasets due to differing grids) and scaled freeze/thaw flags (10 – unfrozen, 20- partially frozen, 30 – frozen, 0 - no data).

**Table 3.** Summary of benchmarking results considering the target transition periods

| Frozen | Thaw | Unfrozen | Freeze-up | frozen |
|---|---|---|---|---|
| winter issue SMAP | Start: best represented in MEaSUREs<br><br>End: partially represented in SMOS | no issues | Start: partially represented in SMAP and MEaSUREs<br><br>End: represented by SMAP | winter issue SMAP |

**Table 4.** Fusion scheme recommendation for northern hemisphere. For product details see Table 1. In case of AM/PM data availability, only AM data to be considered.

| Period defined by day of year | <210 | 210 - 300 | >300 |
|---|---|---|---|
| **Input products** | MEaSUREs (SSMI, AMSR) SMOS | MEaSUREs (SSMI, AMSR) SMAP SMOS | SMOS MEaSUREs (SSMI, AMSR) |
| **Condition** | (1) if one product indicates 'unfrozen' or 'partially frozen' then 'partially frozen', (2) if both indicate 'unfrozen' then 'unfrozen' (3) if both indicate 'frozen' then 'frozen' | (1) if only SMAP 'frozen' then 'partially frozen', (2) if SMAP 'frozen' and ((SSMI or AMSR or SMOS) 'frozen') or (SMOS 'partially frozen')) then 'frozen' (3) if all 'unfrozen' then 'unfrozen' | either use of SSMI or AMSR or SMOS |

and during the thaw period, differentiation needs to be made whether the different products agree with each other or not. SMAP is recommended to be considered for the definition of 'partially frozen' during freeze-up, specifically when its flags disagree with the other products.

The PMW products were fused following the rules defined in Table 4. SMOS and MEaSUREs are recommended to be 375 considered in all three time periods. The SMOS grid cell definition was used as basis (see also figure 2). The previously introduced scheme for flag values (10 – unfrozen, 20 - partially frozen, 30 – frozen) was applied.

### 4.5    Assessment of fused records

Records were fused for the same grid cells as for the initial frozen fraction comparison. The assessment procedure followed the same procedure in a first step. The frozen fraction from Sentinel-1 (SMOS grid) was compared by flag and separate for the 380 thaw and freeze-up period.

In addition, the 'partially frozen' flag has been assessed over three selected borehole locations. The frozen fraction for days flagged in the SMOS product were compared to the fused dataset in order to evaluate the added value of MEaSUREs flags. The 'frozen' and 'unfrozen' flag of the fused product was evaluated considering soil temperature from boreholes at two sites. Results based on reanalyses air temperature data (ERA5) representing an average over a larger area are provided in addition.

**4.5.1    Alaskan North Slope**

The combination of the different products specifically improves the determination of the 'partially frozen' flag for the spring transition (Figure 9). No day with partially frozen conditions was contained in the SMOS product from April to June between

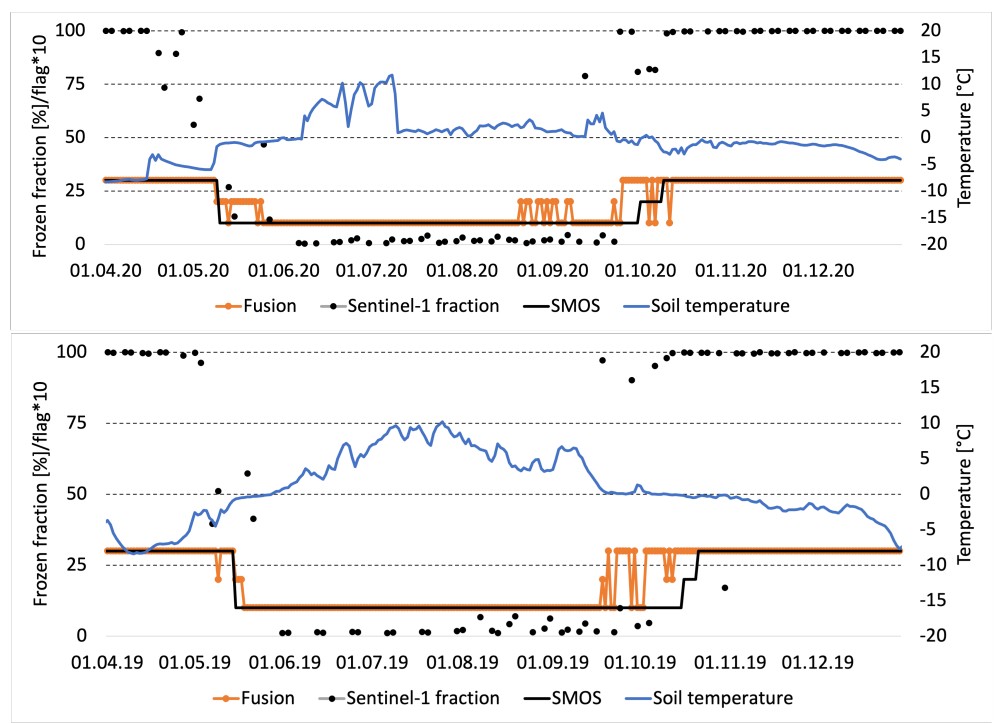

**Figure 9.** Time series example (spring and autumn) of the fused records for the SMOS grid point overlapping with the Happy Valley borehole location for 2020 (top) and the Sagwon borehole location for 2019 (bottom). Near surface soil temperature, Sentinel-1 frozen fraction and scaled freeze/thaw flags for SMOS and Fusion (10 – unfrozen, 20 - partially frozen, 30 – frozen).

2017 and 2020. This gap can be filled by fusion with the MEaSUREs datasets. The majority of grid cells was frozen for days indicated as partially frozen in the SMOS product for August to November for the same years. The fraction distribution is inverted through the fusion with SMAO and MEaSUREs (Figure 11). The average frozen fraction for days defined as 'partially frozen' for SMOS is >90% (Table 5). The Sentinel-1 frozen fraction on days with the 'partially frozen' flag after fusion reduces on average to values between 40% and 46 % at the test sites in Alaska (taking spring and autumn into account, Table ). Frozen fraction values are higher in spring than in autumn (Figure 11). The occurrence of the 'partially frozen' flag in spring increased through the fusion. For example, unfrozen conditions were identified in both MEaSUREs records at the North Slope site on the 24th of May 2018 (Figure 10). The SMOS scheme identified frozen conditions. Part of the grid cells were frozen according to Sentinel-1. The fusion result was 'partially frozen'.

The agreement with near-surface soil temperature records and ERA5 air temperature for the frozen and unfrozen flags was similar for the MEaSUREs, SMOS and Fusion datasets. In general values were higher for ERA5 (as data represent a grid average), and also similar between sites in this case (appr. 90%, Table 6). The agreement with soil temperature differs between sites and was lower (70-80%). Only SMAP showed a lower agreement for the sites on the Alaskan North Slope, with values ranging between 51 and 72%.

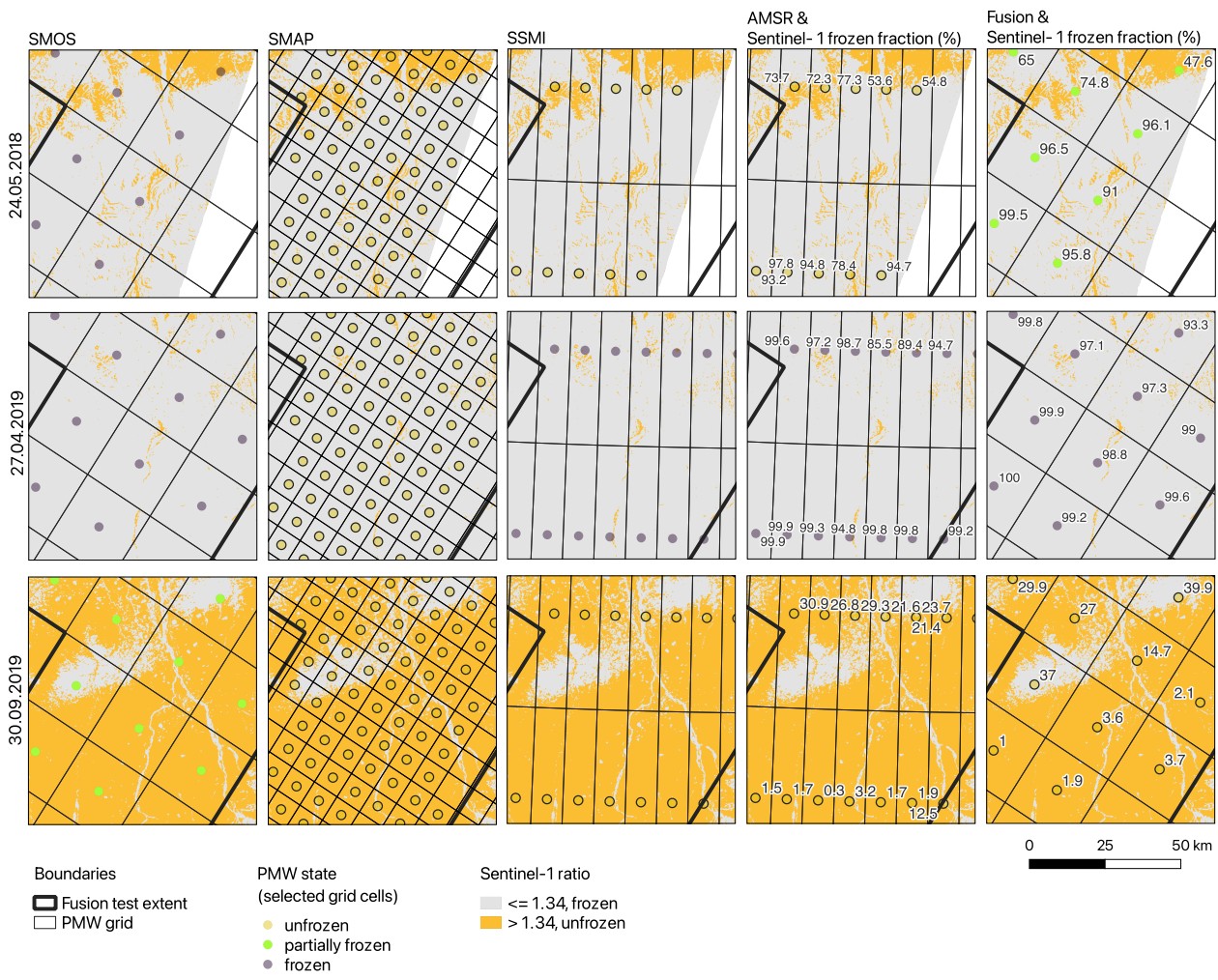

**Figure 10.** Example maps of Sentinel-1 and passive microwave (PMW) surface state, including fusion results for the 24th of May 2018, 27th of April 2019 and 30th of September 2019 (Alaskan North Slope site). For location see Figure 2.

**Table 5.** Average Sentinel-1 derived frozen fraction (in %) for footprints of SMOS and the Fusion freeze/thaw state – days with 'partially frozen' flag only. Sample period 2017 to 2020, three sites on Alaskan North Slope. For analyses extent see Figure 2. For product details see Table 1.

| Site | SMOS | Fusion |
|---|---|---|
| Sagwon | 92 | 46 |
| Happy Valley | 91 | 45 |
| Franklin Bluffs | 91 | 40 |

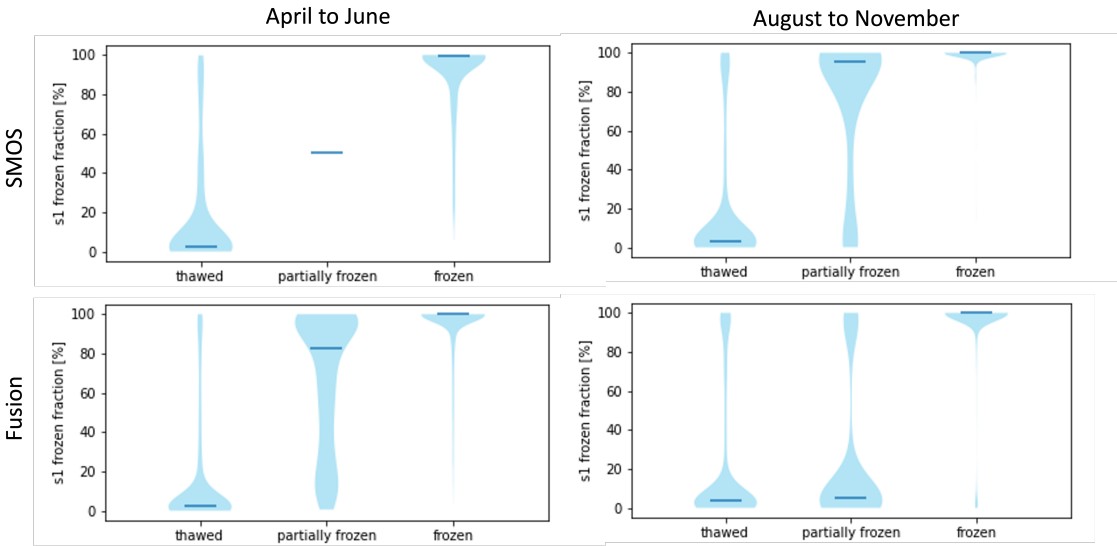

**Figure 11.** Sentinel-1 (s1) derived frozen fraction for footprints of SMOS and the Fusion freeze/thaw state. Sample period 2017 to 2020, Alaskan North Slope. For analyses extent see Figure 2.

**Table 6.** Agreement (in %) of in-situ soil temperature and ERA5 air temperature derived freeze/thaw states for all products and the Fusion result (gridded to SMOS) – days with 'unfrozen' or 'frozen' flag only. Sample period 2017 to 2020, two sites on Alaskan North Slope. For location of sites see Figure 2. For product details see Table 1.

| Site | Type | SSMI | SMOS | SMAP | AMSR | Fusion |
|------|------|------|------|------|------|--------|
| Sagwon | Soil (in-situ) | 82.6 | 84.6 | 60.4 | 83.2 | 83.2 |
|  | Air (ERA5) | 92.4 | 89.2 | 70.6 | 92.3 | 92.7 |
| Happy Valley | Soil (in-situ) | 71.8 | 72.8 | 51.8 | 72.0 | 72.1 |
|  | Air (ERA5) | 92.1 | 88.3 | 70.1 | 90.7 | 92.8 |

### 4.5.2 Finland

Partially frozen conditions are also not captured in the SMOS product for April to June at the Northern Finland site when only considering the in-situ overlap period 2016 to 2018 (Figure 12) at a site located in a depression (at 337m, with altitude ranging from 200 to more than 400 m in the surroundings). An extensive spring thaw period is identified with the Fusion product in Northern Finland starting early May 2017 (Figure 13). In-situ data from under the snowpack (thick enough to insulate the ground with temperatures remaining at approximately 0°C during the winter and as long as the snow remains) indicate that this period corresponds to snowmelt, instead of near surface soil thaw. The extended period of partially frozen conditions also disagrees with the frozen fraction from Sentinel-1 (Figure 13). SMOS as well as AMSR show a high average frozen fraction with 'thawed' (unfrozen) flag over all analysed grid cells (>20%, Figure 12). This also indicates a too early thaw in this region.

AMSR data was used for the fusion what, however, shifts some of these dates to partially frozen (Figure 13). AMSR remained unfrozen until the start of the drop of Sentinel-1 frozen fraction in 2017 while SMOS indicated unfrozen conditions since the beginning of May. This disagreement results in 'partially frozen' following the proposed fusion scheme. The end of this period coincided with ERA5 temperature increase above 0°C. The actual end of thaw for the entire grid cell was, however, one month

later in early July.

The Fusion result example in Figure 13 also includes occurrence of mid-winter FT (source: Bartsch et al. (2023b)). The detected events (15.12.2016, 13.02.2017) correspond to periods with ERA5 temperature above 0°C. All analysed PMW products indicate frozen conditions for those dates.

Frozen fraction values on 'partially frozen' days are high in spring as well as autumn which differs from the Alaskan North Slope results. But they are lower than in the SMOS product. The fusion reduces the high frozen fraction detection for the 'thawed' state for both, spring and autumn.

## 5    Discussion

### 5.1    Benchmark dataset

The use of combined ratios of backscatter intensity using VV and VH polarization as previously suggested for wet snow detection provides a means of reducing the incidence angle effects, but the impact of temperature variations on backscatter intensity at VV during frozen conditions remains (Figure 3). The combined ratio $R_C$ also shows this linkage. The potential influence of temperature on FT retrieval at C-VV has been previously discussed (Naeimi et al., 2012; Bergstedt et al., 2018; Bartsch et al., 2023a). Our results demonstrate that this is specific for VV, but not present in VH (Figure 3). The retrieval

based on VH alone would, however, require an appropriate approach regarding the influence of the incidence angle, such as normalization of $\sigma_0$ (Widhalm et al., 2018) or retrieval of $\gamma_0$ (Small, 2011). An implementation based on VH would also require a location specific threshold determination (instead of global threshold) which is not feasible for the purpose of this study and potential regional application of the FT detection approach.

    The combined ratio approach does nevertheless provide good quality results. The overall agreement with in-situ data of 92%

at the site in Northern Finland is similar to the previously reported accuracy of 94% using the location (pixel) specific threshold determination approach (Bergstedt et al., 2020b). The validation was carried out over a different region (Northern Finland) than the calibration (Alaskan North Slope) confirming the transferability of the method. The performance is also similar to a CNN (Convolutional Neural Networks) approach using both polarization bands (88%, calibration and validation over same region in NE Canada, Chen et al. (2024)).


    We tested the combined ratio approach only over tundra. It can be expected that the performance is lower over forested regions as it relies on the use of VH. A method as suggested by Cohen et al. (2021) using VV only might perform better for

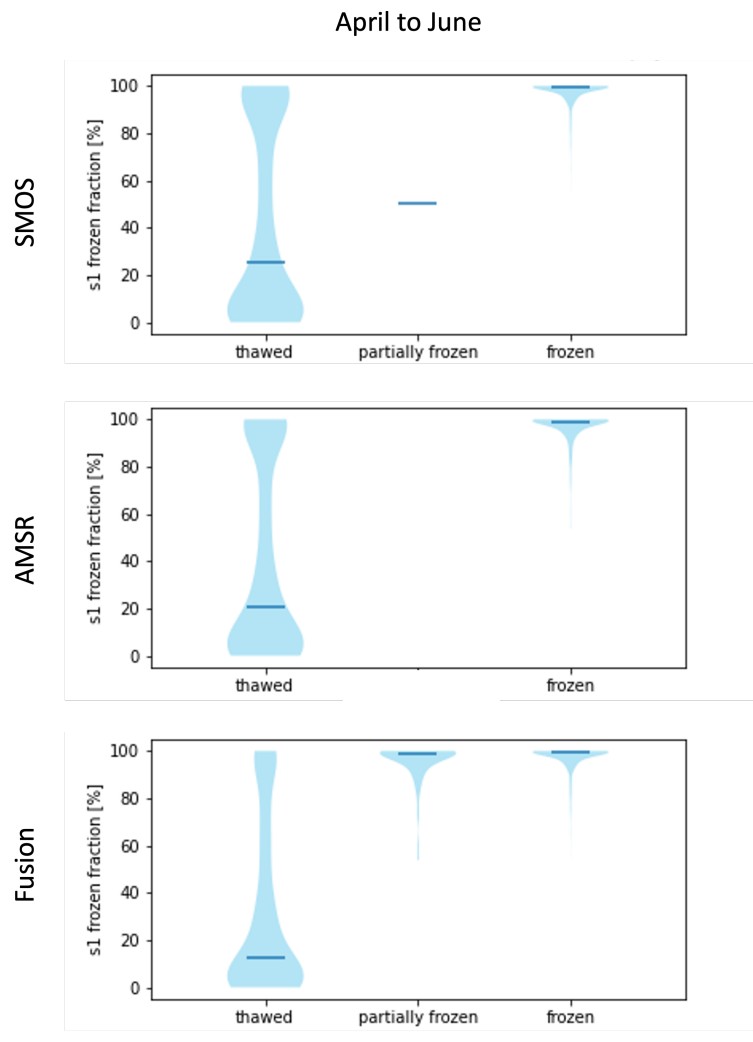

**Figure 12.** Sentinel-1 derived frozen fraction for the Fusion freeze/thaw state (footprints according to SMOS product), Kaldoaivi, Northern Finland. Sample period 2016 to 2018. For analyses area and extent see Figure 2.

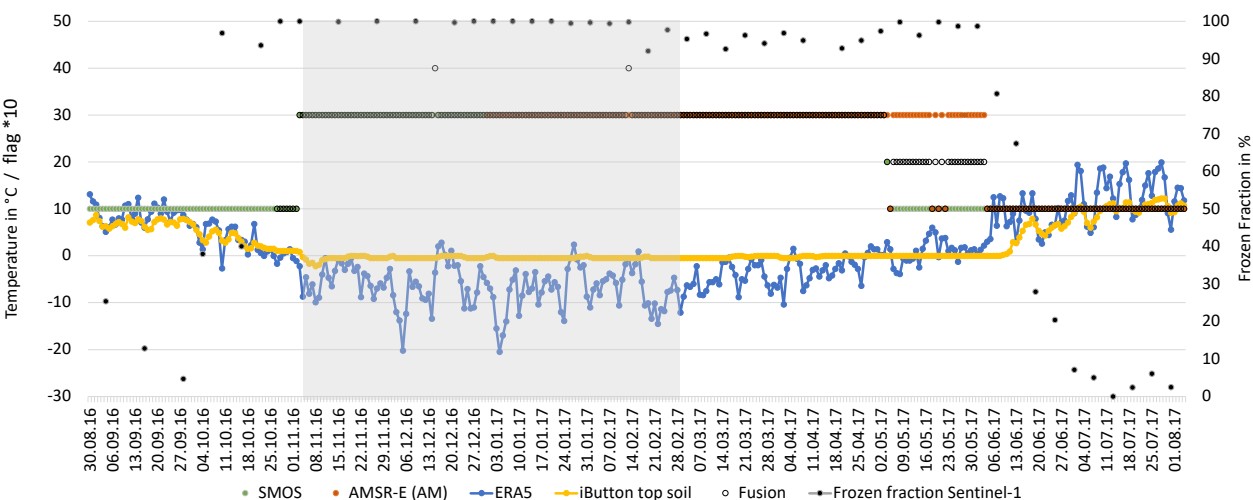

**Figure 13.** Comparison for SMOS grid point overlapping with the Northern Finland site for autumn 2016 and spring 2017 (altitude range 200-400m). Near surface soil temperature (iButton location A4 at 337 m altitude, Bergstedt et al. (2020b)), ERA5 temperature, Sentinel-1 frozen fraction, scaled freeze/thaw flags for SMOS, AMSR and Fusion (10 – unfrozen, 20- partially frozen, 30 – frozen, 40 – mid winter thaw and refreeze). The shaded area indicates the analyses period for mid-winter thaw and refreeze (source: Bartsch et al. (2023b)).

forested areas.

 **5.2 Global/northern hemisphere products benchmarking**

The SMOS - Sentinel-1 comparison results agree with findings of Cohen et al. (2021). Offsets in the FT timing compared to in-situ measurements were described and attributed to the ability of L-band to represent larger soil depths. SMAP which is also based on L-band, does however, represent the end of the autumn freeze-up period well. The AMSR/SSMI frozen flag indicates also a later freeze-up in some cases (multiple switching from frozen/unfrozen, Figure 8), although a much smaller wavelength is used.

Differences in wavelength/frequency (see Table 1) are expected to contribute to the deviations between the results in addition to differences in FT classification method although all depend on thresholding. The MEaSUREs scheme relies on the use of one polarization only with at the same time comparably small wavelength/high frequency. Both SMOS and SMAP FT products are based on the use of the NPR which uses two polarizations but at comparably large wavelength/small frequency. Whereas a single threshold is used to separate thawed from frozen for SMAP, two thresholds are used to derive partially frozen conditions in case of SMOS. The thawed fraction derived from Sentinel-1 is, however, rather low in most cases of 'partially frozen' condition (Table 5). This applies specifically for the spring transition (Figure 7). Once SMOS has indicated a frozen soil condition, the mid-winter values are forced to remain frozen under cold air temperature conditions (Rautiainen et al., 2016;

Rautiainen and Holmberg, 2023) what may contribute to the better agreement with the benchmarking dataset during this period than for SMAP. Both SMOS and SMAP can be both affected by RFI. The SMOS quality flags indicate a potential impact for the Finland site but not for Northern Alaska. The different results for Alaska as shown in the example of Figure 10 may thus originate from the threshold definitions and the consideration of a single channel (V) under certain conditions in case of SMAP. The latter was developed for low latitudes according according to the product documentation (Xu et al., 2023).

SSMI and AMSR records showed very similar performance (e.g. Table 6). The AMSR time series as part of the MEaSUREs dataset is also available at 6 km nominal resolution. The use of this version might be of benefit, replacing both the 25km AMSR and SSMI records. Further analyses would be, however, required for the quantification of the added value of the improved spatial resolution.

The comparison with the Sentinel-1 frozen fraction and in-situ data with the seasonal freeze/thaw prototypes indicates an 470 improvement through the fusion of the different products. A range of issues which differ between the evaluation sites (continuous over Alaska versus sporadic permafrost for Northern Finland), however, remain. Neither the actual end of thaw in spring nor the start of freeze-up in autumn can be captured. These issues cannot be solved based on the existing passive microwave based CDRs.

The frozen fraction derived from Sentinel-1 only applies to areas without open water and barren surfaces. The deviation to the PMW products could therefore also be analysed with respect to open water fraction as has previously investigated for active microwave freeze/thaw Bergstedt et al. (2020a) in order to provide more insight into product issues. An offset in freeze/thaw timing can be expected related to lake ice thaw and formation.

Previous analyses of C-band scatterometer indicated applicability of active microwave data for FT retrieval in high latitude regions (e.g. Naeimi et al. (2012)). Specifically the fusion of SAR and scatterometer is promising for characterization of the transition period (actual determination of start and end, Bergstedt et al. (2020b)). The new SAR based method which has been developed in our study enables operational retrieval of surface state from sensors such as Sentinel-1 and theoretically allows to meet the requirements for a multipurpose FT product if sufficient acquisitions (daily) are available. In addition to soil FT, wet 485 snow detection could be also addressed as it is based on the same pre-processing scheme. The availability of Sentinel-1 data is however constrained and the polarization differs across the Arctic. The required data are unavailable for Greenland and the Canadian High Arctic. In the remaining regions, repeat intervals are not dense enough (daily needed). Bergstedt et al. (2020b) demonstrated, however, the use of historic (one year) Sentinel-1 data for the calibration of a freeze/thaw fraction derived from Metop ASCAT. FT monitoring considering fraction could therefore be implemented in regions with VV/VH availability. Re-490 trieval with HH/HV remains to be tested. An implementation based on Metop ASCAT would allow the production of a CDR going back to 2012 (Metop A and B availability).

More advanced FT algorithms using dielectric models to represent non-linear behavior in highly organic soils common in the Arctic may be able to detect changes in the amount unfrozen liquid water remaining in soil under sub-zero temperatures,

rather than relying on simple binary FT thresholds (e.g. Wang et al. (2024)). Another advanced method employed by Holmberg et al. (2024) shows potential for directly retrieving continuous soil permittivity data over cold regions, currently demonstrated over a test site.

The current benchmarking exercise was limited to AM data in case of MEaSUREs and SMAP due to the availability of Sentinel-1 at the selected sites during this daytime. Diurnal thaw and refreeze is however common (e.g. Bartsch et al. (2007); Böttcher et al. (2018)) during transition periods. The added value of PM data should be addressed in future studies considering other sites or further relevant SAR missions. The future L-band mission NISAR (Das et al., 2021; Rosen and Kumar, 2021) can be expected to be of high value for such benchmarking, as it represents a similar frequency as used for SMOS and SMAP and comparably high temporal sampling will be available. Our analyses is limited to non-forested areas due to the use of C-band SAR. L-band SAR provides the possibility to extent into forests.

## 5.3 Global/northern hemisphere products utility

Current FT products aim at the identification of an average surface state condition within a footprint (Kim et al. (2014) using SSMI, Naeimi et al. (2012) using ASCAT, Derksen et al. (2017) using SMAP and Rautiainen et al. (2016) using SMOS), but specific applications including permafrost, soil moisture and greenhouse gas fluxes require information on the start of thaw/freeze-up and completion of thaw/freeze-up (Tables B1, B2, B3Bartsch et al. (2022).

### 5.3.1 Permafrost

The observation that the unfrozen period (complete thaw of grid cell) in high latitude relatively cold regions is longer in case of SSMI than suggested by the benchmarking data (Figure 8) agrees with findings of Kroisleitner et al. (2018). The unfrozen period was also found longer than for an experimental product based on Metop ASCAT. This impedes the applicability for permafrost related applications such as the estimation of potential mean annual ground temperature. The impact on the capability to monitor long-term trends in frozen period length (as suggested by Park et al. (2016a)) remains to be investigated.

Kouki et al. (2019) also compared and evaluated datasets for the start of spring thaw. SSMI/SMMR switch from frozen to thawed showed to correspond to the start of increase of temperatures above 0°C what agrees with the Sentinel-1 fraction and in-situ borehole comparison on the Alaskan North Slope (Figure 8). Results from northern Finland also indicate that the frozen/thaw switch (start of decrease of frozen fraction) coincides with increasing air temperatures, but not with upper soil temperatures (Figure 13). Snow is still present on the ground at this time as indicated through the temperature measurements (remains stable close to 0°C) and in-situ measurements of snow depth in March 2018 (44 to 86 cm, Bergstedt and Bartsch (2020)).

A combination of MEaSUREs (SSMI/AMSR) and/or SMOS with SMAP (excluding the winter period) may fulfill the requirements for a freeze/thaw flag use as proxy for potential permafrost occurrence. Threshold requirements (Table B1) could at least be partially met. Diurnal variations may need to be considered with respect to acquisition timing. Freeze/thaw cycles on a daily basis have been shown common during the spring period in the context of wet snow detection (e.g. Bartsch et al., 2007). This pattern may extent into the snow free period.

Ground temperatures can also be affected by melt of snow in mid-winter (Westermann et al., 2011). Figure 13) provides an example of an existing microwave remote sensing based product and the co-occurrence of air temperature increase. As near surface soil temperatures were already at 0°C in this zone with only sporadic permafrost. A temperature response was thus not detected. Previous studies including Westermann et al. (2011) have, however, shown increases in colder regions. A combined product is therefore expected to be of benefit for permafrost applications.

### 5.3.2 Masking of soil moisture products

With the currently available FT products, target requirements for soil moisture (completely unfrozen determination, Table B2) cannot be met for the transition periods. A partial solution provides the combination of SMOS and MEaSUREs for spring thaw. The combination improved the partially frozen flag of SMOS on the Alaskan North Slope (Table 5). Precise determination of start of freeze-up in Autumn based on the existing products is currently not possible.

Results of Bergstedt et al. (2020b) indicate that Metop ASCAT backscatter in combination with Sentinel-1 frozen fraction statistics allows to derive start and end of the thaw and freeze transition. A calibration would, however, require the processing of Sentinel-1 for at least one year for each ASCAT footprint to be processed. The frozen fraction could be similarly used for combination with the MEaSUREs and SMOS data for the spring period and for defining the start of freeze-up which is not represented yet.

### 5.3.3 Vegetation and carbon flux applications

The combination of MEaSUREs, SMOS and SMAP is expected to partially fulfill threshold requirements for the classification (Table B3), as the start and end of the transition periods are of relevance in this case. To fulfill the target requirements, also the start of freeze-up would be needed. The consideration of Sentinel-1 facilitated frozen fraction retrieval would be of benefit. AM and PM information would be eventually required in addition to indication of presence of melting snow in order to monitor diurnal variations which are of relevance for fluxes.

## 6 Conclusions

The joint use of VV and VH available from Sentinel-1 acquisitions has been shown applicable for FT retrieval in high latitudes. This allows for regional scale analyses at comparably high spatial resolution as a global threshold can be defined. The creation

of a freeze/thaw benchmarking dataset requires sufficient temporal sampling at the same time. Both spatial coverage and temporal sampling are major constraints, but implementation based on Sentinel-1 A and B is possible for some regions in the Arctic including Alaska and northern Scandinavia. The resulting benchmarking dataset provides a means to address issues in coarse resolution satellite products from PMW data across the Arctic although representing a different frequency. Current quality assessment with in-situ data is limited due to the scarce availability of ground stations and their limited representativeness because of high landscape heterogeneity.

The thresholding and further paramterization applied to the two L-band records, SMAP and SMOS, result in substantial differences across all seasons between the products despite of similar input. This might be specific to tundra environment. Further investigations are needed to identify actual error sources and to revise the fusion scheme for the identification of 'partially frozen' conditions.

For global and northern hemisphere applications coarser resolution datasets such as the tested passive microwave based datasets need to be used at this stage. Improvements of the existing regional to global datasets are, however, needed. The transition periods are not well captured even when a 'partially frozen' flag, as in case of SMOS, is considered. Such a flag should be (1) included in any future coarse spatial resolution FT product and (2) improved in precision in order to meet various user requirements. A fusion with SAR retrievals may allow to overcome various error sources, specifically seasonally changing water surfaces, which are an issue due to the coarse spatial resolution of PMW observations. The utility of FT datasets could be further enhanced by combination with wet snow products from direct observations or indirect via snow structure change resulting from refreezing snow.

A fusion of the existing products can only partially enhance the accuracy. The Sentinel-1 derived benchmark dataset provides a means for improving existing retrieval schemes and the development of new products that so far relied on comparisons with in-situ point or modelled and coarser resolution reanalyses data. Comparison to in-situ records also need to consider soil temperature measurements in order to evaluate the role of snow.

*Author contributions.* AB developed the concept for the study, analysed the results and wrote the first draft of the manuscript. XM, MH and KR have processed the satellite data. JW, TN and KR contributed to the conception of the study and writing of the manuscript. HB and DN contributed to the in-situ surveys, their compilation and to the writing of the manuscript.

*Competing interests.* The authors declare no competing interests.

*Acknowledgements.* This work was supported by the European Space Agency CCI+ Permafrost and CCI+ Snow and AMPAC-Net projects. DN acknowledges the US NSF grant 1832238 for supporting his involvement in this study.

**Appendix A: SMAP data quality example**

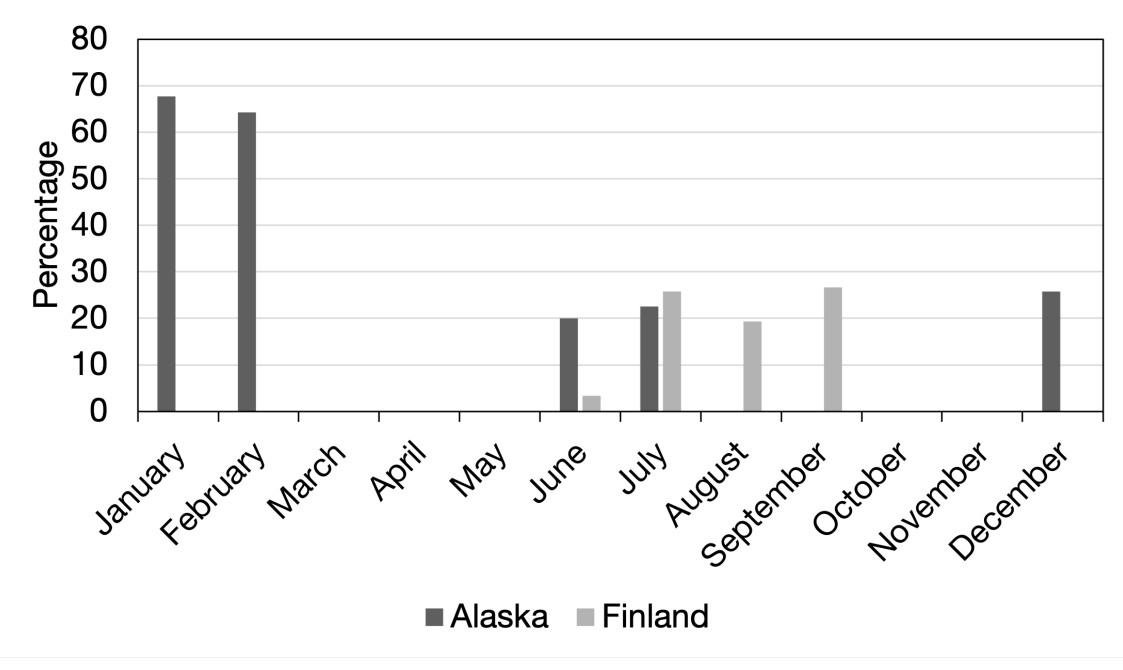

**Figure A1.** Examples of SMAP quality flag for use of AMSR-E or brightness temperature mitigation (Xu et al., 2023) in 2021 (percentage per month).

## Appendix B: User requirements summary from Bartsch et al. (2022)

**Table B1.** Requirements for a FT climate data record for permafrost monitoring in lowland areas in line with Permafrost_cci (Bartsch et al., 2023e).

|  | Threshold requirement | Target requirement |
|---|---|---|
|  | Coverage and sampling | |
| **Geographical coverage** | Pan-Arctic | Global with regional specific products |
| **Temporal sampling** | Annually (aggregated from daily data) | Daily |
| **Temporal extent** | Last decade | 1979 - present |
|  | Thematic content | |
| **Target classes** | Frozen, unfrozen | Frozen, unfrozen, melting |
| **Accuracy** | Better accuracy than available so far | <20 % error |
|  | Resolution | |
| **Horizontal resolution** | 10 km | 100m-1km |
| **Subgrid variability** | no | yes |

**Table B2.** Requirements for a FT climate data record for masking of soil moisture (National Research Council, 2014; Dunbar, 2018; Entekhabi et al., 2014; Trofaier et al., 2017).

|  | Threshold requirement | Target requirement |
|---|---|---|
|  | Coverage and sampling | |
| **Geographical coverage** | North of 45°N | Global |
| **Temporal sampling** | As soil moisture product | Daily |
| **Temporal extent** | As length of specific mission | 1979 - present |
|  | Thematic content | |
| **Target classes** | Frozen, unfrozen | Frozen, partially frozen, unfrozen (complete) |
| **Accuracy** | Better accuracy than available so far | <20 % error |
|  | Resolution | |
| **Horizontal resolution** | As soil moisture product | As soil moisture product |
| **Subgrid variability** | no | yes |

**Table B3.** Requirements for a FT climate data record for vegetation and carbon flux applications (Böttcher et al., 2018; Bartsch et al., 2007; Aalto et al., 2020).

| | Threshold requirement | Target requirement |
|---|---|---|
| | **Coverage and sampling** | |
| **Geographical coverage** | Northern hemisphere | Global |
| **Temporal sampling** | Annually (aggregated from daily data) | Diurnal |
| **Temporal extent** | Last decade | 1979 - present |
| | **Thematic content** | |
| **Target classes** | Start and end day of year of unfrozen period | Frozen, partially frozen, unfrozen (complete), melting snow |
| **Accuracy** | 7.5 days for date of freeze-up, less than 10% of days with missing data | 5 days for date of freeze-up |
| | **Resolution** | |
| **Horizontal resolution** | 1° | 100m-25 km |
| **Subgrid variability** | yes | yes |

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
