# Peer review of "Benchmarking passive microwave satellite derived freeze/thaw datasets"

_EGUsphere, 2024_

## Author Response (AR1)

Dear editor,

We have revised the manuscript according to all reviewer comments.

**We would like to thank both reviewers for their very helpful and detailed revisions and comments.**

**Reviewer 1**

The authors state that a target requirement for a FT product domain aiming at permafrost should be global (Ln 67-69). However, permafrost and FT events generally only occur in the high latitudes and high altitude regions. As such, more specifics and justification on the global domain requirement is needed here.

- Old: The target should be a global product
  New: The target should be a global product including not only the Northern Hemisphere but also mountain ranges and polar ice free regions on the southern hemisphere.

Ln 125: It should be noted that the MEaSUREs FT record is based on classification of Ka-band (V-Pol.) brightness temperature records from various PMW satellite missions (SMMR, SSM/I, AMSR). Clarification of the frequency of the FT retrieval is important given the strong link between frequency and the characteristic penetration and sensitivity of the retrieval to different landscape features (e.g. vegetation, snow, soil). Further, the MEaSUREs record includes calibrated brightness temperatures from the two AMSR missions (AMSR-E [2003-2011] and AMSR2 [2012-present]). This should be noted and the more general "AMSR" term used in place of "AMSR-E" throughout (e.g. Table 1 and elsewhere).
- Reply: we added the frequency information and precise sensor naming throughout the manuscript

Table 1: Include additional information in the table summary stating the underlying frequency represented from the different FT products used given expected frequency dependent differences in sensitivity and performance. Also, indicate whether global or polar grid datasets are used since some products (e.g. MEaSUREs, SMAP) provide both.
- Reply: The projections are SMAP - Equal-Area Scalable Earth Grid 2.0 (EASE-Grid 2.0) - Northern Hemisphere (Lambert Azimuthal Equal Area) at 9km. SMOS - Equal-Area Scalable Earth Grid 2.0 (EASE-Grid 2.0) - Northern Hemisphere (Lambert Azimuthal Equal Area) at 25km. MEaSUREs - NSIDC Equal-Area Scalable Earth (EASE)-Grid 2.0 Global (cylindrical, equal-area) at 25 km. We included this in the table and text.

Section 2: More justification of the particular FT data records and formats selected for this study is needed given the diversity of product formats available, which may be better suited to polar regions. For example, the MEaSUREs archive includes both global 25-km and polar 6-km grid FT products. The SMAP L3FT products (SPL3FTP[E]) include both global and polar projections in both 36-km and 9-km grid formats. The SMAP 9-km grid product is derived using a spatially optimized (Backus-Gilbert) interpolation of SMAP antenna temperatures and associated brightness temperatures intended to improve the effective spatial resolution. Because of the excessive stretching of the global cylindrical equal-area (EASE-

grid) map projections toward the poles (e.g. illustrated by the elongated MEaSUREs grid in Fig 2.) the polar azimuthal EASE-grid products are likely better suited for applications over high latitude tundra, especially involving comparisons with in situ ground stations.

- Reply: the 6km MEaSUREs product only represents AMSR/E. We therefore focused on the 25km only. In case of SMAP, the 9km product was used in polar projection. We added this in the table. We also added a comment on the 6km dataset in the discussion and rephrased the data section for clarification:

  Old: The AMSR-E/AMSR2 based time series integrated in the NASA MEaSUREs program is a 6 km resolution record
  New: The AMSR-E/AMSR2 time series of the MEaSUREs dataset is also available as 6 km resolution record in addition to the 25 km record which includes time series from all sensors. The 6km dataset covers the northern and southern hemisphere currently spanning the period 2002-2020 (Kim2021). However, only the 25 km product was used for this study.
  New in discussion: SSMI and AMSR records showed very similar performance (e.g. Table 6). The AMSR time series as part of the MEaSUREs dataset is also available at 6 km nominal resolution. The use of this version might be of benefit, replacing both the 25km AMSR and SSMI records. Further analyses would be, however, required for the quantification of the added value of the improved spatial resolution.

Figure 2 caption: More information is needed in the figure caption noting the different product grids represented in the image; e.g. polar vs global grid projections; 36-km vs 9-km Res.

- Response: we now refer to the extended table and added: Grids of the passive microwave products (MEaSUREs, SMAP and SMOS) differ in projection and extent. For details see Table 1.

Section 3.1: The authors state that the spatial frozen fraction is derived from the SAR data for each cell of the PMW FT products (e.g. Ln 192). This approach is a major strength of this study for validating coarser resolution land retrievals that may only contain a single representative value for each grid cell and then rely on validation against sparse in situ measurements that may not be fully representative of the complex landscape within the satellite footprint. The use of SAR FT data for benchmarking overcomes the limitations of sparse in situ measurements for being regionally representative and more congruent with the sampling footprint from other global FT records. This challenge is partially depicted by the varying FT grid sizes relative to the point measurements in Figure 2, but a more clear statement of this strength is needed.

- Reply: we agree and added at the end of the first paragraph of the conclusions:
- New: The resulting benchmarking dataset provides a means to address issues in coarse resolution satellite products from PMW data across the Arctic. Current quality assessment with in situ data is limited due to the scarce availability of ground stations and their limited representativeness because of high landscape heterogeneity.

Section 3.1: The PMW FT products (e.g. MEaSUREs, SMAP) include diurnal (AM and PM) FT retrievals from the satellite ascending and descending orbital passes. The authors should

clarify which FT sampling period(s) were used in this study and provide justification for the selections (e.g. temporal optimization with in situ or SAR sampling, best accuracy/ consistency, etc.).

- Reply: AM data was chosen where available since most SAR data were acquired at AM times at the Alaskan benchmarking site (97%). We added this in section 2.1, and switched this section with 2.2. (Sentinel-1).

Ln 250: The authors state that data from in situ measurements were used for assessing FT conditions indicated from the PMW FT product fusion. However, more information is needed describing how the spatial match-ups for the comparisons were made between in situ measurements representing local site conditions and the spatially gridded data, particularly from the SAR; e.g. were comparisons made between the in situ station measurements and the overlying SAR pixel or against a larger SAR subregion within the overlapping PMW grid cells?

- Reply: the SAR comparison was made pixel based. SAR data have been sampled to 100x100m throughout the processing. We added in section 3.2:
  The SAR data have been sampled to 100mx100m.
  And in section 3.3
  Old: Air temperature as well as soil temperature measurements were considered.
  New: Air temperature as well as soil temperature measurements were compared to the Rc value of each corresponding 100mx100m SAR pixel.

Ln 406-409: It should also be noted here that MEaSUREs and SMAP FT performances strongly degrade with increasing fractional water (FW) cover common to tundra; in general the SMAP L-band data appear more strongly impacted than MEaSUREs Ka-band retrievals as documented from Kim et al. (2019). The latest SMAP product release also includes an improved water mask expected to enhance performance particularly in boreal-Arctic areas with abundant water bodies (Du et al. 2023). The PMW FT products include QC flags for FW, which could be used to help inform the stand-alone or Fusion approaches.

- Reply: The SMAP FW water values are very low for both sites. They are close to 0 for the considered North slope grid points. Values are similar for Northern Finland except one grid point which includes a lake but has less than 5% water fraction. The sites were also selected considering distance to the coastline and resolution of FT products. SMAP flags indicate use of SCV retrieval. Comments have been added in the data section and discussion.

Section 4: The accuracy and utility of the PMW FT and fused products were assessed against the SAR frozen fraction and in situ observations. However, it is unclear whether additional ancillary data quality information provided with the PMW FT products, including pixel level quality (QC) information on expected performance and retrieval success, was used to help interpret and explain product behavior. For example, the MEaSUREs FT product contains annual maps of estimated quality along with per pixel QC data on the presence of known error factors including extensive FW, terrain complexity, and active precipitation days. The SMAP FT record includes similar per pixel QC flags. The tundra sites used for this study may be affected by these adverse conditions, which may help explain some of the degraded performance. The authors should include this additional information in the results discussion.

- Reply: Both sites are labeled as high quality areas in the MEaSUREs dataset. SMOS quality flags indicate temporarily up to 5 (out of the 17) cells as potentially affected by RFI over northern Finland. No RFI issues are indicated for the Alaskan sites. It can however occur that there are gaps of up to 7 days between updates at all sites. SMAP flags indicate retrieval with SCV correlation <5% and the use of AMSR-E or TB mitigation in some cases. These cases occurred in mid-summer and mid-winter. For Alaska this includes mostly end of December to end of March, and in some cases also July and August. For Finland they occur from July to September (where freeze-up starts in October). The misclassification in SMAP in winter does however occur over both sites.

Figures 8, 9: It was unclear from the methods description whether the AM or PM overpass derived FT records were used for these comparisons. As noted in the results, some of the records have trouble fully delineating the transitional periods, but I wonder whether the transition would be better resolved using both the AM and PM FT series, as we might expect greater frequency of diurnal thawing and refreezing during these periods.
- Reply: for the purpose of benchmarking we used AM only, see above. But we agree that considering PM may bring advance regarding user needs. We added in the discussion:
- The current benchmarking exercise was limited to AM data in case of MEASURES and SMAP due to the availability of Sentinel-1 during this daytime. Diurnal thaw and refreeze is however common (e.g. Bartsch et al. 2007, Böttcher et al. 2018) during transition periods. The added value of PM data should be addressed in future studies considering further relevant SAR missions. The future L-band mission NISAR (Das et al. 2021, Rosen et al. 2021) can be expected to be of high value for such benchmarking, as it represents a similar frequency as used for SMOS and SMAP and comparably high temporal sampling will be available. Our analyses is limited to non-forested areas due to the use of C-band SAR. L-band SAR provides the possibility to extent into forests.

Section 5.3: More discussion is needed here regarding the potential utility of FT retrievals from different microwave frequencies for the desired application. In this study, C-band SAR data are used with in situ ground temperature observations to benchmark a fusion of FT products derived from Ka-, and L-band PMW observations. However, this approach neglects the frequency dependence of the FT products, which may represent different and potentially complementary aspects of FT heterogeneity from different landscape features (e.g. vegetation vs soil). The relative utility of FT products for the different applications described (Vegetation, soil moisture, etc..) is also expected to have strong frequency dependence. Given this dependence, is the use of the frozen area reference from C-band SAR data a constraint in benchmarking FT performance from different frequency retrievals? A more clear and concise statement of the study strengths and limitations in this regard is needed.
- Reply: we have partially discussed the issue of vegetation at the end of section 5.2 referring to problems of C-band in forested regions. We therefore limited the analyses to none-forest areas. We extended the discussion on L-band as in the response to the question on AM data use above, data quality, methodology and summarized this now also in the discussion and conclusions.
- Discussion: The future L-band mission NISAR (Das et al. 2021, Rosen et al. 2021) can be expected to be of high value for such benchmarking, as it represents a similar frequency as used for SMOS and SMAP and comparably high temporal sampling will be available. Our

analyses is limited to non-forested areas due to the use of C-band SAR. L-band SAR provides the possibility to extent into forests.

- Conclusions: The resulting benchmarking dataset provides a means to address issues in coarse resolution satellite products from PMW data across the Arctic although representing a different frequency. Current quality assessment with in-situ data is limited due to the scarce availability of ground stations and their limited representativeness because of high landscape heterogeneity.

Section 5.3 Cont.: In this study the frozen fraction reference is defined in a spatial context and the FT condition is defined in accordance with binary FT states defined from temperature measurements. The simple binary FT classification approach likely contributes to frequency dependent differences in the PMW, SAR and site level FT comparisons. This is a constraint common to the FT products used, which should be mentioned.

> Reply: we added the following sentences at the end of the first paragraph of section 5.2, after "… although a much smaller wavelength is used." referring to new text in the data description as suggested by referee 2:
> New:
> • Differences in wavelength/frequency (see Table 1) are expected to contribute to the deviations between the results in addition to differences in FT classification method although all depend on thresholding. The MEASUREs scheme relies on the use of one polarization only with at the same time comparably small wavelength/high frequency. Both SMOS and SMAP FT products are based on the use of the NPR which uses two polarizations but at comparably large wavelength/small frequency. Whereas a single threshold is used to separate thawed from frozen for SMAP, two thresholds are used to derive partially frozen conditions in case of SMOS. The thawed fraction derived from Sentinel-1 is however rather low in most cases of 'partially frozen' condition (Table 5). This applies specifically for the spring transition (Figure 7).
> • Once SMOS has indicated a frozen soil condition, the mid-winter values are forced to remain frozen under cold air temperature conditions (Rautiainen et al. 2016, ATBD: Rautiainen et al. 2023, https://earth.esa.int/documents/d/earth-online/smos-soil-freeze-and-thaw-state-atbd) what may contribute to the better agreement with the benchmarking dataset during this period than for SMAP.

-

Further, more sophisticated FT algorithms may be able to detect changes in the amount unfrozen liquid water remaining in soil under sub-zero temperatures, rather than relying on simple binary FT thresholds (e.g. Wang et al. 2024). This requires advanced dielectric models to represent non-linear behavior in highly organic soils common in the arctic, but is more physically meaningful and may enhance application utility, which may be worth considering in future FT benchmarking efforts.

- Reply: we have added: More advanced FT algorithms using dielectric models to represent non-linear behavior in highly organic soils common in the Arctic may be able to detect changes in the amount unfrozen liquid water remaining in soil under sub-zero temperatures, rather than relying on simple binary FT thresholds (e.g. Wang et al. 2024). Another advanced method employed by Holmberg et al. (2024) shows potential for directly retrieving continuous soil permittivity data over cold regions, currently demonstrated over a test site.

Minor comments:

Changed as suggested:
      Ln 20: "as proxy" should be "as a proxy".
      Ln 86: "this not" should be "this is not".
      Ln 191: "fulfil" should be "fulfill".
      Ln 144: Include Derksen et al. 2017 or Kim et al. 2019 (below) as more suitable citations on the stated SMAP accuracy relevant to this study since the Kraatz paper doesn't involve tundra or other high northern latitude sites.
      Ln 273: "higher then" should be "higher than".
      Table 4 and Figure 11: "AMSR-E" should be restated more broadly as "AMSR" to encompass both AMSR-E and AMSR2 portions of the FT record.
      Ln 368: "what differs" should be "which differs".
      Ln 436-437: "SSMR" should be "SMMR"; "what agrees" should be "which agrees".
      Ln 444: "fulfil" should be "fulfill"
      Ln 471: "relevenace" should be "relevance"; also clarify what is meant by "fluxes" here; e.g. previous studies have used AM and PM FT retrievals to characterize transient nighttime frost events affecting vegetation growth and day-night FT transitions commonly occurring in the shoulder seasons.
      Ln 476: "constrains" should be "constraints".

Ln 325: Sentence needs revising for better clarity.

- Reply: Thanks for spotting.
- Old: The PMW were fused following the rules defined in commended to be considered in all three time periods and using theSMOS grid cell definition
- New: The PMW products were fused following the rules defined in Table 4. SMOS MEaSUREs are recommended to be considered in all three time periods and the use of the SMOS grid cell definition.

**Reviewer 2**

This paper used a fusion approach to detect soil FT status over Alaska North slope and Northern Finland using three FT data records. The results demonstrated the combined ratio approach provides excellent overall agreement with in-situ measurements and a similar performance to a CNN approach. The results also show the potential capability of benchmarking PMW FT data products with Sentinel-1 FT retrievals. The paper covers a topic that is suitable to readers of The Cryosphere and should be of particular interest to those interested in FT classification algorithm development over the Cryosphere and permafrost monitoring under climate change. The suggested major revisions are as follows:

- Major concern is lack of detail in describing FT data fusion method and benchmark FT classification algorithm. Some references were included in this manuscript; however, it is not

sufficient description for readers to get clear and repeat the FT data retrieval processing. For example, spatial and temporal resolutions for fused and benchmark FT data sets are not clearly described in the method section.

- Reply: please see our suggestions below
- Fusion: The fused record is daily as for the input datasets. Spatial resolution corresponds to SMOS.
  - Old: All comparisons of the Fusion dataset were made on the basis of the SMOS grid as the new grid.
  - New: The fusion of the daily records was made on the basis of the SMOS grid (25km). All comparisons of the Fusion dataset were also made on the basis of the SMOS grid.
- Benchmark dataset details:
  - Old: The time period covered for Finland corresponds with the in-situ data availability (August 2016 to August 2018, acquisitions every 12 days). Data processed for Alaska span three years from 2018 to 2020 (on average an acquisition every 9 days using overlapping orbits).
  - New: The time period covered for Finland corresponds with the in-situ data availability (August 2016 to August 2018, acquisitions every 12 days). Data processed for Alaska span three years from 2018 to 2020 (on average an acquisition every 9 days using overlapping orbits). This results in differing sampling intervals between the regions and lower sampling compared to the global FT records.
  - The SAR data have been sampled to 100x100m

- In the accuracy agreement, authors show the temporal comparison with in-situ soil temperatures. However, differences in spatial FT patterns among FT products (with different frequency, resolution, and so on) would allow us to better understand sub-scale impacts on FT classification accuracy. Comparison of S1 frozen fraction maps and other FT state images (e.g., fused datasets) would be good to analyze their spatial patterns. Providing a couple of sample images would be enough.

- Reply: Example maps for spring have been added.
- New text: The occurrence of the 'partially frozen' flag in spring increased through the fusion. For example, unfrozen conditions were identified in both MEaSUREs records at the North Slope site on the 24th of May 2018 (Figure 10). The SMOS scheme identified frozen conditions. Part of the grid cells were frozen according to Sentinel-1. The fusion result was 'partially frozen'.

- In Table 6, why does accuracy differs among the operational coarse resolution products (MEASURES, SMAP, SMOS)? Did you use high-quality FT flags? Several studies reported some possible reasons why there are differences particularly during the transitional periods. Authors should include the potential reasons briefly in the discussion section (e.g., different sensor specification) even though some of them are stated in the introduction section. If any unknown uncertainties are found in this study, you would include some challenges and probably suggest future study in the conclusion section to improve fused FT data products (stated shortly only in Line 479, page 25).

> Reply: the selected regions have good quality according to the flags provided. See also response to reviewer 1 comments. RFI might lead to issues for some of the grid point

in Northern Finland. Details have been added in the evaluation method section. And a comment on RFI was added to the discussion.

Additional edits are noted below:

Line 128, page 5: Did you use MEaSUREs FT data Version 5 in this study? Version 3 was introduced in line 124 , page 4 as well. Authors should clarify it.
- Reply: Yes, we used v5. Added to Table 1

Line 126 & 132, page 5; Which frequency and what algorithm was used for MEASURES FT datasets? Authors should include it.
- Reply: We added this information to Table 1 and the following passages to the text:

    The different sensors represent different frequencies (see Table 1).

    The MEaSUREs algorithm is based on a seasonal threshold approach. The threshold was derived annually on a grid-cell-wise basis (Kim et al. 2017) using an empirical relationship between brightness temperature and daily surface air temperature records from global model reanalysis. The SMMR-SSM/I-SSMIS record was developed by merging the Scanning Multichannel Microwave Radiometer (SMMR), Special Sensor Microwave Imager (SSM/I), and Special Sensor Microwave Imager/Sounder (SSMIS) 37 GHz frequency (vertical polarization) brightness temperature records (Kim et al. 2014). The AMSR-E/AMSR2 records represent 36 GHz.

    SMAP FT state was determined used a seasonal threshold approach (Derksen et al. 2017). A normalized polarization ratio (using V and H brightness temperature; 1.41 GHz) is assessed. A winter and summer reference is derived from frozen and thawed soil conditions. It is calculated for each year and averaged over the entire SMAP period (ATBD, Xu et al. 2020). These values are used for derivation of the threshold and a seasonal scale factor.

    Thaw and freeze references are also used in case of SMOS FT retrieval (Rautiainen et al. 2016, ATBD: Rautiainen et al. 2023). For FT state estimation, the normalized polarization ratio (NPR, 1.400 GHz–1.427 GHz) values are scaled using the reference NPR values from frozen and thawed soil conditions. All potential observations from the frozen soil and thawed soil conditions are collected from the winter and summer periods when reanalysis air temperature data has indicated <-3C and >+3C, respectively. From these collected potential values, the 50 most extreme ones are stored and their median used as a reference value. These are pixel-wise values. Then two thresholds are used to differentiate between three FT states from the scaled NPR.

Ine 131, page 5: Table 1 does not include a 6km resolution FT data sets. Authors should modify it.
- Reply: we only included used records. See also our suggestion to modify the text above.
- Old: The AMSR-E/AMSR2 based time series integrated in the NASA MEaSUREs program is a 6 km resolution record

- New: The AMSR-E/AMSR2 time series of the MEaSUREs dataset is also available as 6 km resolution record in addition to the 25 km record which includes time series from all sensors. Only the latter was used for this study

-

Line 132, Page 5: The AMSR-E/AMSR2 data is available between 2002 and 2021 (not 2020).
- changed

Line 136, page 5: what FT algorithm was used for SMAP and SMOS products? Authors should describe briefly.
- Added, see above

Line 145, page 5: How did authors determine the partially frozen state from SMOS? Does partially frozen mean thawing in Table 1? Is it thaw-refreeze condition? Is it different from SMAP transitional state?
- Reply: It differs from the SMAP transitional state which is based on differences between AM and PM. SMOS partial thaw is derived from a scaled NPR & reference value indicator. See also our suggestions for text edits above.

Line 170-171, page 6: Authors should include the temperature data source (e.g., web link, references).
- Reply: added now in the text and Table 2

Line 182, page 6: Authors should include more details on ERA5 reanalysis (e.g., data source, spatial and temporal resolutions, which temperature?, soil or air temperature? and so on).
- Reply: the details are - Copernicus Atmosphere Monitoring Service information, 0.1° grid, hourly (daily average derived where no time stamp), air temperature. It is only used for (1) visualization in one figure (12) to highlight the difference between soil and air temperature and (2) complementing in situ validation (Table 6). We suggest changed the order of listing in Table 6 (first soil and then ERA5) and clarified this in the text at several places.

Line 192, page 8: It is not clear to me how to derive the frozen fraction from Sentinel-1. Authors should specify it.
- Reply: we modified section 3.4 (line 192 is part of the general workflow section 3.1):
- Old: The frozen/unfrozen fraction was derived for each of the overlapping footprints.
- New: The frozen/unfrozen fraction was derived for each of the overlapping footprints of the PMW products (grid cells as shown in Figure 2). The proportion of unmasked Sentinel-1 pixels classified as frozen has been determined for each acquisition.

Line 203-204, page 9: The equation, Rc should be included here to better understand.
- added

Line 211, page 9: This Rc equation should be introduced first in Line 203-204.
- adjusted

Line 216, page 9: which reanalysis data? And which temperature was used? Is air or soil temperature? Daily or hourly?

- Naeimi et al. (2012): ERA-Interim soil temperature, 0.7° grid, daily values derived from 00, 06, 12, 18 data; for validation GLDAS-Noah 0.25° grid, daily derived from 3-hourly data, near surface soil
- Bergstedt et al. (2020b): no reanalyses used. We rephrasied for clarification:
- This was based on reanalyses data in Naemi et al. (2012). Bergstedt et al. (2020b) tested calibration based on in situ air temperature over a limited number of sites and validated the results with in situ near surface soil temperature data. Here we also used in situ data, air and soil, but from a larger dataset and also for calibration.

Line 248, page 10: what are the other flags? Authors should specify it.
- Reply: 'thawed' and 'frozen'

Line 254-255, page 10: is it daily mean air temperature?
- Reply: daily mean for soil and air temperature, added

Line 257, page 10: what is the fusion method? How to fuse Metop ASCAT and SMOS?
- Reply: The result from Metop ASCAT (rain-on-snow event yes/now) was postprocessed based on SMOS NPR analyses (wet snow condition within a specific period before and after the event). We suggest to rephrase:
- Old: As an example, mid-winter thaw and refreeze as available from fusion of Metop ASCAT and SMOS
- New: As an example, mid-winter thaw and refreeze as available from combination of Metop ASCAT and SMOS

Line429, Page24: It (Finding … 2018) is not clearly stated. Authors should clarify it.
- Old: Findings for SSMI agree with Kroisleitner et al. (2018). This applies specifically to the observation that the unfrozen period (complete thaw of grid) in high latitude relatively cold regions is longer than suggested by the benchmarking data (this study, Figure 8) and than detected in the experimental product based on Metop ASCAT (Kroisleitner et al., 2018).
- New: The observation that the unfrozen period (complete thaw of grid) in high latitude relatively cold regions is longer in case of SSMI than suggested by the benchmarking data (Figure 8) agrees with findings of Kroisleitner et al. (2018). The unfrozen period was also found longer than for an experimental product based on Metop ASCAT.

Line488,page 26: situ should be in-situ.
- changed

Figure 1: Legend describing permafrost zones (types) should be included inside the map, even though the permafrost zones are described in the caption.
- added

Figure 2: It seems to me that MEaSUREs grids (in yellow) may not be 25km resolution. The length of horizontal lines looks like ~9km.

Reply: The grid outlines correspond to the "25km" product grid in global EASE Grid (cylindrical equal-area) projection visualized at UTM zone 6N. It does not translate to 25 x 25 km squares. At the specific latitude it translates to 10km x 62.5km (which results in the same area covered). We added this together with the projection information to Table 1 and related text.

Figure 6 and 7: what do horizontal blue lines mean?
- Reply: average, added to captions

Table 6: ERA5 air or soil temperature?
- Reply: air temperature, added.